# How to Turn Your Knowledge Graph Embeddings into Generative Models

**Lorenzo Loconte**
University of Edinburgh, UK
`l.loconte@sms.ed.ac.uk`

**Nicola Di Mauro**
University of Bari, Italy
`nicola.dimauro@uniba.it`

**Robert Peharz**
TU Graz, Austria
`robert.peharz@tugraz.at`

**Antonio Vergari**
University of Edinburgh, UK
`avergari@ed.ac.uk`

## Abstract

Some of the most successful knowledge graph embedding (KGE) models for link prediction – CP, RESCAL, TUCKER, COMPLEX – can be interpreted as energy-based models. Under this perspective they are not amenable for exact maximum-likelihood estimation (MLE), sampling and struggle to integrate logical constraints. This work re-interprets the score functions of these KGEs as *circuits* – constrained computational graphs allowing efficient marginalisation. Then, we design two recipes to obtain efficient generative circuit models by either restricting their activations to be non-negative or squaring their outputs. Our interpretation comes with little or no loss of performance for link prediction, while the circuits framework unlocks exact learning by MLE, efficient sampling of new triples, and guarantee that logical constraints are satisfied by design. Furthermore, our models scale more gracefully than the original KGEs on graphs with millions of entities.

## 1 Introduction

Knowledge graphs (KGs) are a popular way to represent structured domain information as directed graphs encoded as collections of triples (subject, predicate, object), where subjects and objects (entities) are nodes in the graph, and predicates their edge labels. For example, the information that the drug "loxoprofen" interacts with the protein "COX2" is represented as a triple (loxoprofen, interacts, COX2) in the biomedical KG ogbl-biokg [32]. As real-world KGs are often incomplete, we are interested in performing reasoning tasks over them while dealing with missing information. The simplest reasoning task is *link prediction* [48], i.e., querying for the entities that are connected in a KG by an edge labelled with a certain predicate. For instance, we can retrieve all proteins that the drug "loxoprofen" interacts with by asking the query (loxoprofen, interacts, ?).

Knowledge graph embedding (KGE) models are state-of-the-art (SOTA) models for link prediction [40, 56, 12] that map entities and predicates to sub-symbolic representations, which are used to assign a real-valued degree of existence to triples in order to rank them. For example, the SOTA KGE model COMPLEX [65] assigns (loxoprofen, interacts, phosphoric-acid) and (loxoprofen, interacts, COX2) scores 2.3 and 1.3, hence ranking the first higher than the second in our link prediction example.

This simple example, however, also highlights some opportunities that are missed by KGE models. First, triple scores cannot be directly compared across different queries and across different KGE models, as they can be seen as *negative energies* and not *normalised probabilities over triples* [41, 7, 6, 27, 43]. To establish a sound probabilistic interpretation and therefore have *probabilities instead of scores* that can be easily interpreted and compared [79], we would need to compute the normalisation

37th Conference on Neural Information Processing Systems (NeurIPS 2023).

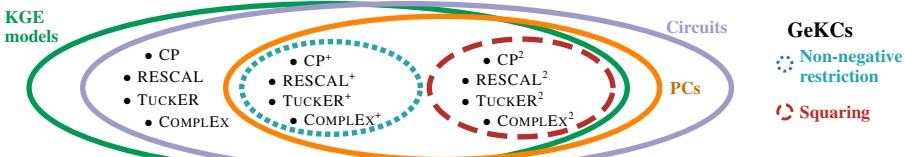

Figure 1: **Which KGE models can be used as efficient generative models of triples?** The score functions of popular KGE models such as COMPLEX, CP, RESCAL and TUCKER can be easily represented as circuits (lilac). However, to retrieve a valid probabilistic circuit (PC, in orange) that encodes a probability distribution over triples (GeKCs) we need to either *restrict its activations to be non-negative* (in blue, see Section 4.1) or *square it* (in red, see Section 4.2).

constant (or partition function), which is impractical for real-world KGs due to their considerable size (see Section 2). Therefore, learning KGE models by maximum-likelihood estimation (MLE) would be computationally infeasible, which is the canonical probabilistic way to learn *a generative model over triples*. A generative model would enable us to sample new triples efficiently, e.g., to generate a surrogate KG whose statistics are consistent with the original one or to do data augmentation [9]. Furthermore, traditional KGE models do not provide a principled way to *guarantee the satisfaction of hard constraints*, which are crucial to ensure trustworthy predictions in safety-critical contexts such as biomedical applications. The result is that predictions of these models can blatantly violate simple constraints such as KG schema definitions. For instance, the triple that the SOTA COMPLEX ranks higher in our example above violates the semantics of "interacts", i.e., such predicate can only hold between drugs (e.g., loxoprofen) and proteins (e.g., COX2) but phosphoric-acid is not a protein.

**Contributions.** We show that KGE models that have become a de facto standard, such as COMPLEX and alternatives based on multilinear score functions [47, 40, 66], can be represented as structured computational graphs, named *circuits* [13]. Under this light, **i)** we propose a different interpretation of these computational graphs and their parameters, to retrieve efficient and yet expressive probabilistic models over triples in a KG, which we name *generative KGE circuits* (GeKCs) (Section 4). We show that **ii)** not only GeKCs can be efficiently trained by exact MLE, but learning them with widely-used discriminative objectives [37, 40, 56, 12] also scales far better over large KGs with millions of entities. In addition, **iii)** we are able to sample new triples exactly and efficiently from GeKCs, and propose a novel metric to measure their quality (Section 7.3). Furthermore, by leveraging recent theoretical advances in representing circuits [76], **iv)** we guarantee that predictions at test time will never violate logical constraints such as domain schema definitions by design (Section 5). Finally, our experimental results show that these advantages come with no or minimal loss in terms of link prediction accuracy.

## 2 From KGs and embedding models...

**KGs and embedding models.** A KG $\mathcal{G}$ is a directed multi-graph where nodes are entities and edges are labelled with predicates, i.e., elements of two sets $\mathcal{E}$ and $\mathcal{R}$, respectively. We define $\mathcal{G}$ as a collection of triples $(s, r, o) \subseteq \mathcal{E} \times \mathcal{R} \times \mathcal{E}$, where $s$, $r$, $o$ denote the subject, predicate and object, respectively. A *KG embedding* (KGE) model maps a triple $(s, r, o)$ to a real scalar via a *score function* $\phi \colon \mathcal{E} \times \mathcal{R} \times \mathcal{E} \to \mathbb{R}$. A common recipe to construct differentiable score functions for many state-of-the-art KGE models [56] is to (i) map entities and predicates to *embeddings* of rank $d$, i.e., elements of normed vector spaces (e.g., $\mathbb{R}^d$), and (ii) combine the embeddings of subject, predicate and object via multilinear maps. This is the case for KGE models such as CP [40], RESCAL [47], TUCKER [3], and COMPLEX [65] (see Fig. 2). For instance, the score function of COMPLEX [65] is defined as $\phi_{\text{COMPLEX}}(s, r, o) = \text{Re}(\langle \mathbf{e}_s, \mathbf{w}_r, \overline{\mathbf{e}_o} \rangle)$ where $\mathbf{e}_s, \mathbf{w}_r, \mathbf{e}_o \in \mathbb{C}^d$ are the complex embeddings of subject, predicate and object, $\langle \cdot, \cdot, \cdot \rangle$ denotes a trilinear product, $\overline{\cdot}$ denotes the complex conjugate operator and $\text{Re}(\cdot)$ the real part of complex numbers.

**Probabilistic loss-derived interpretation.** KGE models have been traditionally interpreted as *energy-based models* (EBMs) [41, 7, 6, 27, 43]: their score function is assumed to compute the negative energy of a triple. This interpretation induces a distribution over possible KGs by associating a Bernoulli variable, whose parameter is determined by the score function, to every triple [48]. Learning EBMs under this perspective requires using contrastive objectives [7, 48, 56], but several recent works observed that to achieve SOTA link prediction results one needs only to predict

subjects, objects [37, 40, 56] or more recently also predicates of triples [12], i.e., to treat KGEs as *discriminative* classifiers. Specifically, they are learned by minimising a categorical cross-entropy, e.g., by maximising $\log p(o \mid s, r) = \phi(s, r, o) - \log \sum_{o' \in \mathcal{E}} \exp \phi(s, r, o')$ for object prediction. From this perspective, we observe that we can recover an energy-based interpretation if we assume there exist a joint probability distribution $p$ over three variables $S, R, O$, denoting respectively subjects, predicates and objects. Written as a Boltzmann distribution, we have that $p(S = s, R = r, O = o) = (\exp \phi(s, r, o))/Z$, where $Z = \sum_{s' \in \mathcal{E}} \sum_{r' \in \mathcal{R}} \sum_{o' \in \mathcal{E}} \exp \phi(s', r', o')$ denotes the partition function. This interpretation is apparently in contrast with the traditional one over possible KGs [48]. We reconcile it with ours in Appendix E. Under this view, we can reinterpret and generalise the recently introduced discriminative objectives [12] as a weighted *pseudo-log-likelihood* (PLL) [70]

$$\mathcal{L}_{\text{PLL}} := \sum_{(s,r,o) \in \mathcal{G}} \omega_s \log p(s \mid r, o) + \omega_o \log p(o \mid s, r) + \omega_r \log p(r \mid s, o) \tag{1}$$

where $\omega_s, \omega_o, \omega_r \in \mathbb{R}_+$ can differently weigh each term, which is a conditional log-probability that can be computed by summing out either $s$, $r$ or $o$, e.g., to compute $\log p(o \mid s, r)$ above. Optimisation is usually carried out by mini-batch gradient ascent [56] and, given a batch of triples $B \subset \mathcal{G}$, we have that exactly computing the PLL objective requires time $\mathcal{O}(|\mathcal{E}| \cdot |B| \cdot \text{cost}(\phi))$ and space $\mathcal{O}(|\mathcal{E}| \cdot |B|)$ to exploit GPU parallelism [36],[1] where $\text{cost}(\phi)$ denotes the complexity of evaluating the $\phi$ once.

Note that the PLL objective (Eq. (1)) is a traditional proxy for learning *generative* models for which it is infeasible to evaluate the *maximum-likelihood estimation* (MLE) objective[2]

$$\mathcal{L}_{\text{MLE}} := \sum_{(s,r,o) \in \mathcal{G}} \log p(s, r, o) = -|\mathcal{G}| \log Z + \sum_{(s,r,o) \in \mathcal{G}} \phi(s, r, o). \tag{2}$$

In theory, evaluating $\log p(s, r, o)$ exactly can be done in polynomial time under our three-variable interpretation, as computing $Z$ requires $\mathcal{O}(|\mathcal{E}|^2 \cdot |\mathcal{R}| \cdot \text{cost}(\phi))$ time, but in practice this cost is still prohibitive for real-world KGs. In fact, it would require summing over $|\mathcal{E} \times \mathcal{R} \times \mathcal{E}|$ evaluations of the score function $\phi$, which for FB15k-237 [62], the small fragment of Freebase [48], translates to ~$10^{11}$ evaluations of $\phi$. This practical bottleneck hinders the generative capabilities of these models and their ability to yield normalised and interpretable probabilities. Next, we show how we can reinterpret KGE score functions as to retrieve a generative model over triples for which computing $Z$ exactly can be done in time $\mathcal{O}((|\mathcal{E}| + |\mathcal{R}|) \cdot \text{cost}(\phi))$, making renormalisation feasible.

## 3   ...to Circuits...

In this section, we show that popular and successful KGE models such as CP, RESCAL, TUCKER and COMPLEX (see Fig. 2 and Section 2), can be viewed as structured computational graphs that can, in principle, enable summing over all possible triples efficiently. Later, to exploit this efficient summation for marginalisation over triple probabilities, we reinterpret the semantics of these computational graphs as to yield circuits that output valid probabilities. We start with the needed background about circuits and show that some score functions can be readily represented as circuits.

**Definition 1** (Circuit [13, 76]). A *circuit* $\phi$ is a parametrized computational graph over variables $\mathbf{X}$ encoding a function $\phi(\mathbf{X})$ and comprising three kinds of computational units: *input*, *product*, and *sum*. Each product or sum unit $n$ receives as inputs the outputs of other units, denoted with the set $\text{in}(n)$. Each unit $n$ encodes a function $\phi_n$ defined as: (i) $l_n(\text{sc}(n))$ if $n$ is an input unit, where $l_n$ is a function over variables $\text{sc}(n) \subseteq \mathbf{X}$, called its *scope*, (ii) $\prod_{i \in \text{in}(n)} \phi_i(\text{sc}(\phi_i))$ if $n$ is a product unit, and (iii) $\sum_{i \in \text{in}(n)} \theta_i \phi_i(\text{sc}(\phi_i))$ if $n$ is a sum unit, with $\theta_i \in \mathbb{R}$ denoting the weighted sum parameters. The scope of a product or sum unit $n$ is the union of the scopes of its inputs, i.e., $\text{sc}(n) = \bigcup_{i \in \text{in}(n)} \text{sc}(i)$.

Fig. 2 and Fig. A.1 show examples of circuits. Next, we introduce the two structural properties that enable efficient summation and Prop. 1 certifies that the aforementioned KGEs have these properties.

**Definition 2** (Smoothness and Decomposability). A circuit is *smooth* if for every sum unit $n$, its input units depend all on the same variables, i.e, $\forall i, j \in \text{in}(n): \text{sc}(i) = \text{sc}(j)$. A circuit is *decomposable* if the inputs of every product unit $n$ depend on disjoint sets of variables, i.e, $\forall i, j \in \text{in}(n)$ $i \neq j: \text{sc}(i) \cap \text{sc}(j) = \varnothing$.

---

[1]For large real-world KGs it is reasonable to assume that $|\mathcal{E}| \gg |\mathcal{R}|$.

[2]In general, the PLL is recognised as a proxy for MLE because, under certain assumptions, it is possible to show it can retrieve the MLE solution asymptotically with enough data [35].

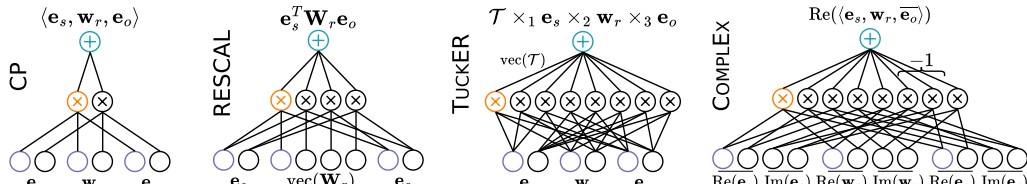

Figure 2: **Interpreting the score functions of CP, RESCAL, TUCkER, COMPLEX as circuits** over 2-dimensional embeddings. Input, product and sum units are coloured in purple, orange and blue, respectively. Output sum units are labelled with the score functions, and their parameters are assumed to be 1, if not specified. The detailed construction is presented in Appendix A.1. Given a triple $(s, r, o)$, the input units map subject $s$, predicate $r$ and object $o$ to their embedding entries. Then, the products are evaluated before the weighted sum, which outputs the score of the input triple.

**Proposition 1** (Score functions of KGE models as circuits)**.** The computational graphs of the score functions $\phi$ of CP, RESCAL, TUCkER and COMPLEX are smooth and decomposable circuits over $\mathbf{X} = \{S, R, O\}$, whose evaluation cost is $\mathrm{cost}(\phi) \in \Theta(|\phi|)$, where $|\phi|$ denotes the number of edges in the circuit, also called its size. For example, the size of the circuit for CP is $|\phi_{\mathrm{CP}}| \in \mathcal{O}(d)$.

Appendix A.1 reports the complete proof by construction for the score functions of these models and the circuit sizes, while Fig. 2 illustrates them. Intuitively, since the score functions of the cited KGE models are based on products and sums as operators, they can be represented as circuits where input units map entities and predicates into the corresponding embedding components (similarly to look-up tables). As the inputs of each sum unit are product units that share the same scope $\{S, R, O\}$ and fully decompose it in their input units, they satisfy smoothness and decomposability (Def. 2).

Smooth and decomposable circuits enable summing over all possible (partial) assignments to $\mathbf{X}$ by (i) performing summations at input units over values in their domains, and (ii) evaluating the circuit once in a feed-forward way [13, 76]. This re-interpretation of score functions allows to "push" summations to the input units of a circuit, greatly reducing complexity, as detailed in the following proposition.

**Proposition 2** (Efficient summations)**.** Let $\phi$ be a smooth and decomposable circuit over $\mathbf{X} = \{S, R, O\}$ that encodes the score function of a KGE model. The sum $\sum_{s \in \mathcal{E}} \sum_{r \in \mathcal{R}} \sum_{o \in \mathcal{E}} \phi(s, r, o)$ or any other summation over subjects, predicates or objects can be computed in time $\mathcal{O}((|\mathcal{E}| + |\mathcal{R}|) \cdot |\phi|)$.

However, *these sums are in logarithm space*, as we have that $p(s, r, o) \propto \exp \phi(s, r, o)$ (see Section 2). As a consequence, summation in this context *does not* correspond to marginalising variables in probability space. This drives us to reinterpret the semantics of these circuit structures as to operate directly in probability space, rather than in logarithm space, i.e., by encoding non-negative functions.

# 4  ... to Probabilistic Circuits

We now present how to reinterpret the semantics of the computational graphs of KGE score functions to directly output non-negative values for any input. That is, we cast them as *probabilistic circuits* (PCs) [13, 76, 19]. First, we define our subclass of PCs that encodes a possibly unnormalised probability distribution over triples in a KG, but allows for efficient marginalisation.

**Definition 3** (Generative KGE circuit)**.** A *generative KGE circuit* (GeKC) is a smooth and decomposable PC $\phi_{\mathsf{pc}}$ over variables $\mathbf{X} = \{S, R, O\}$ that encodes a probability distribution over triples, i.e., $\phi_{\mathsf{pc}}(s, r, o) \propto p(s, r, o)$ for any $(s, r, o) \in \mathcal{E} \times \mathcal{R} \times \mathcal{E}$.

Since our GeKCs are smooth and decomposable (Def. 2) they guarantee the efficient computation of $Z$ in time $\mathcal{O}((|\mathcal{E}| + |\mathcal{R}|) \cdot |\phi_{\mathsf{pc}}|)$ (Prop. 2). This is in contrast with existing KGE models, for which it would require the evaluation of the whole score function on each possible triple (Section 2). For example, assume a non-negative CP score function $\phi_{\mathrm{CP}}^+(s, r, o) = \langle \mathbf{e}_s, \mathbf{w}_r, \mathbf{e}_o \rangle \in \mathbb{R}_+$ for some embeddings $\mathbf{e}_s, \mathbf{w}_r, \mathbf{e}_o \in \mathbb{R}^d$. Then, we can compute $Z$ by pushing the outer summations inside the trilinear product, i.e., $Z = \langle \sum_{s \in \mathcal{E}} \mathbf{e}_s, \sum_{r \in \mathcal{R}} \mathbf{w}_r, \sum_{o \in \mathcal{E}} \mathbf{e}_o \rangle$, which can be done in time $\mathcal{O}((|\mathcal{E}| + |\mathcal{R}|) \cdot d)$. In the following sections, we propose two ways to turn the computational graphs of CP, RESCAL, TUCkER and COMPLEX into GeKCs without additional space requirements.

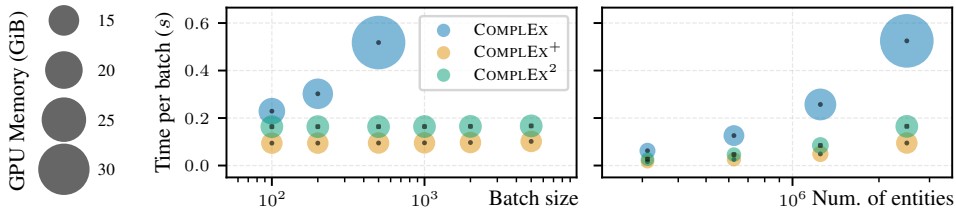

Figure 3: **GeKCs scale better.** Time (in seconds) and peak GPU memory (in GiB as bubble sizes) required for computing the PLL objective and back-propagating through it for a single batch on ogbl-wikikg2, by increasing the batch size and number of entities. See Appendix C.4.3 for details.

## 4.1 Non-negative restriction of a score function

The most natural way of casting existing KGE models to GeKCs is by constraining the computational units of their circuit structures (Section 3) to output non-negative values only. We will refer to this conversion method as the *non-negative restriction* of a model. To achieve the non-negative restriction of CP, RESCAL and TUCKER we can simply restrict the embedding values and additional parameters in their score functions to be non-negative, as products and sums are operators closed in $\mathbb{R}_+$. Thus, each input unit $n$ over variables $S$ or $O$ (resp. $R$) in a non-negative GeKC encodes a function $l_n$ (Def. 1) modelling an unnormalised categorical distribution over entities (resp. predicates). For example, each $i$-th entry $e_{si}$ of the embedding $\mathbf{e}_s \in \mathbb{R}_+^d$ associated to an entity $s \in \mathcal{E}$ becomes a parameter of the $i$-th unnormalised categorical distribution over $S$. See Fig. C.1 for an example. We denote the non-negative restriction of these KGEs as CP⁺, RESCAL⁺ and TUCKER⁺, respectively.

However, deriving COMPLEX⁺ [65] by restricting the embedding values of COMPLEX to be non-negative is not sufficient, because its score function includes a subtraction as it operates on complex numbers. To overcome this, we re-parameterise the imaginary part of each complex embedding to be always greater than or equal to its real part. Appendix C.2 details this procedure. Even if this reparametrisation allows for more flexibility, imposing non-negativity on GeKCs can restrict their ability to capture intricate interactions over subjects, predicates and objects given a fixed number of learnable parameters [67]. We empirically confirm this in our experiments in Section 7. Therefore, we propose an alternative way of representing KGEs as PCs via *squaring*.

## 4.2 Squaring the score function

Squaring works by taking the score function of a KGE model $\phi$, and multiplying it with itself to obtain $\phi^2 = \phi \cdot \phi$. Note that $\phi^2$ would be guaranteed to be a PC, as it always outputs non-negative values. The challenge is to represent the product of two circuits as a smooth and decomposable PC as to guarantee efficient marginalisation (Prop. 2).[3] In general, this task is known to be #P-hard [76].

However, it can be done efficiently if the two circuits are *compatible* [76], as we further detail in Appendix B.1. Intuitively, the circuit representations of the score functions $\phi$ of CP, RESCAL, TUCKER and COMPLEX (see Fig. 2) are simple enough that every product unit is defined over the same scope $\{S, R, O\}$ and fully decomposes it on its input units. As such, these circuits can be easily multiplied with any other smooth and decomposable circuit, a property also known as *omni-compatibility* [76]. This property enables us to build the squared version of these KGE models, which we denote as CP², RESCAL², TUCKER² and COMPLEX², as PCs that are still smooth and decomposable. Note that these squared GeKCs do allow for negative parameters, and hence can be more expressive. The next theorem, instead, guarantees that we can normalize them efficiently.

**Theorem 1** (Efficient summations on squared GeKCs). *Performing summations as stated in Prop. 2 on CP², RESCAL², TUCKER² and COMPLEX² can be done in time $\mathcal{O}((|\mathcal{E}| + |\mathcal{R}|) \cdot |\phi|^2)$.*

For instance, the partition function $Z$ of CP² with embedding size $d$ would require $\mathcal{O}((|\mathcal{E}| + |\mathcal{R}|) \cdot d^2)$ operations to be computed, while a simple feed-forward pass for a batch of $|B|$ triples is still $\mathcal{O}(|B| \cdot d)$. While in this case marginalisation requires an increase in complexity that is quadratic in $d$, it is still faster than the brute force approach to compute $Z$ (see Section 2 and Appendix C.4.1).

---

[3]In fact, even though $\phi^2$ can be easily built by introducing a product unit over two copies of $\phi$ as sub-circuits, the resulting circuit would be not decomposable (Def. 2), as the sub-circuits are defined on the same scope.

**Quickly distilling KGEs to squared GeKCs.** Consider a squared GeKC obtained by initialising its parameters with those of its energy-based KGE counterpart. If the score function of the original KGE model *always* assigns non-negative scores to triples, then the "distilled" squared GeKC will output the *same exact ranking of the original model for the answers to any link prediction queries*. Although the premise of the non-negativity of the scores might not be guaranteed, we observe that, in practice, learned KGE models do assign positive scores to all or most of the triples of common KGs (see Appendix D). Therefore, we can use this observation to either instantly distil a GeKC or provide a good heuristic to initialise its parameters and fine-tune them (by MLE or PLL maximisation). In both cases, the result is that we can convert the original KGE model into a GeKC that provides comparable probabilities, enable efficient marginalisation, sampling, and the integration of logical constraints with little or no loss of performance for link prediction (Section 7.1).

### 4.3 On the Training Efficiency of GeKCs

GeKCs also offer an unexpected opportunity to better scale the computation of the PLL objective (Eq. (1)) on very large knowledge graphs. This is because computing the PLL for a batch of $|B|$ triples with GeKCs obtained via non-negative restriction and by squaring (Sections 4.1 and 4.2) does not require storing a matrix of size $\mathcal{O}(|\mathcal{E}| \cdot |B|)$ to fully exploit GPU parallelism [36]. For instance, in Appendix C.4.2 we show that computing the PLL for CP [40] with embedding size $d$ requires time $\mathcal{O}(|\mathcal{E}| \cdot |B| \cdot d)$ and additional space $\mathcal{O}(|\mathcal{E}| \cdot |B|)$. On the other hand, for $\text{CP}^2$ (resp. $\text{CP}^+$) it requires time $\mathcal{O}((|\mathcal{E}| + |B|) \cdot d^2)$ (resp. $\mathcal{O}((|\mathcal{E}| + |B|) \cdot d)$) and space $\mathcal{O}(|B| \cdot d)$. Table C.1 summarises similar reduced complexities for other instances of GeKCs, such as $\text{COMPLEX}^+$ and $\text{COMPLEX}^2$. The reduced time and memory requirements with GeKCs allow us to use larger batch sizes and better scale to large knowledge graphs. Fig. 3 clearly highlights this when measuring the time and GPU memory required to train these models on a KG with millions of entities such as ogbl-wikikg2 [32].

### 4.4 Sampling new triples with GeKCs

GeKCs only allowing non-negative parameters, such as $\text{CP}^+$, $\text{RESCAL}^+$ and $\text{TUCKER}^+$, support *ancestral sampling* as sum units can be interpreted as marginalised discrete latent variables, similarly to the latent variable interpretation in mixture models [55, 53] (see Appendix C.3 for details). This is however not possible in general for $\text{COMPLEX}^+$ and GeKCs obtained by squaring, as negative parameters break this interpretation. Luckily, as these circuits still support efficient marginalisation (Prop. 2 and Thm. 1) and hence also conditioning, we can perform *inverse transform sampling*. That is, to generate a triple $(s, r, o)$, we can sample in an autoregressive fashion, e.g., first $s \sim p(S)$, then $r \sim p(R \mid s)$ and $o \sim p(O \mid s, r)$, hence requiring only three marginalization steps.

## 5 Injection of Logical Constraints

Converting KGE models to PCs provides the opportunity to "embed" logical constraints in the neural link predictor such that (i) predictions are always guaranteed to satisfy the constraints at test time, and (ii) training can still be done by efficient MLE (or PLL). This is in stark contrast with previous approaches for KGEs, which relax the constraints or enforce them only at training time (see Section 6). Consider, as an example, the problem of integrating the logical constraints induced by a schema of a KG, i.e., enforcing that triples not satisfying a *domain constraint* have probability zero.

**Definition 4** (Domain constraint). Given a predicate $r \in \mathcal{R}$ and $\kappa_S(r), \kappa_O(r) \subseteq \mathcal{E}$ the sets of all subjects and objects that are semantically coherent with respect to $r$, a *domain constraint* is a propositional logic formula defined as

$$K_r \equiv S \in \kappa_S(r) \wedge R = r \wedge O \in \kappa_O(r) \equiv (\vee_{u \in \kappa_S(r)} S = u) \wedge R = r \wedge (\vee_{v \in \kappa_O(r)} O = v). \quad (3)$$

Given $\mathcal{R} = \{r_1, \ldots, r_m\}$ a set of predicates, the disjunction $K \equiv K_{r_1} \vee \ldots \vee K_{r_m}$ encodes all the domain constraints that are defined in a KG. An input triple $(s, r, o)$ satisfies $K$, written as $(s, r, o) \models K$, if $s \in \kappa_S(r)$ and $o \in \kappa_O(r)$. To design GeKCs such as their predictions always satisfy logical constraints (which might not be necessarily domain constraints), we follow Ahmed et al. [1] and define a score function to represent a probability distribution $p_K$ that assigns probability mass only to triples that satisfy the constraint $K$, i.e., $\phi_{\text{pc}}(s, r, o) \cdot c_K(s, r, o) \propto p_K(s, r, o)$. Here, $\phi_{\text{pc}}$ is a GeKC and $c_K(s, r, o) = \mathbb{1}\{(s, r, o) \models K\}$ is an indicator function that ensures that zero mass is assigned to triples violating $K$. In words, we are "cutting" the support of $\phi_{\text{pc}}$, as illustrated in Fig. 4.

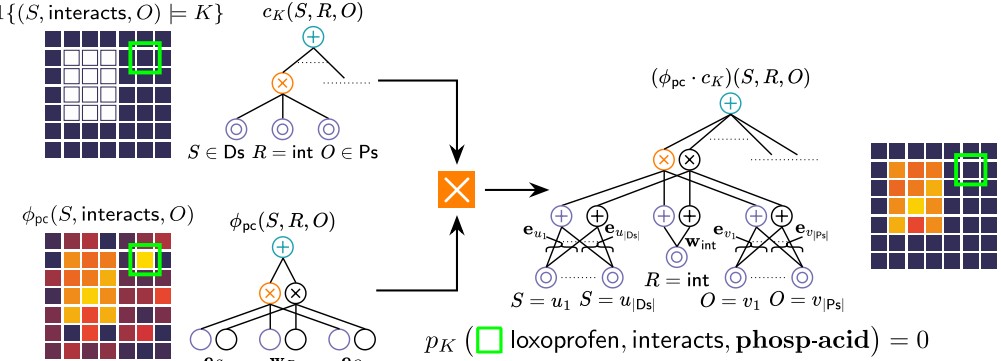

Figure 4: **Injection of domain constraints.** Given a circuit $c_K$ encoding domain constraints and a GeKC $\phi_{\text{pc}}$, the probability assigned by the product circuit $\phi_{\text{pc}} \cdot c_K$ to the inconsistent triple showed in Section 1 is 0, and a positive probability is assigned to consistent triples only, e.g., for the interacts predicate those involving drugs (Ds) as subjects and proteins (Ps) as objects. Best viewed in colours.

Computing $p_K(s, r, o)$ exactly but naively would require computing a new partition function $Z_K = \sum_{s' \in \mathcal{E}} \sum_{r' \in \mathcal{R}} \sum_{o' \in \mathcal{E}} (\phi_{\text{pc}}(s', r', o') \cdot c_K(s', r', o'))$, which is impractical as previously discussed (Section 2). Instead, we compile $c_K$ as a smooth and decomposable circuit, sometimes called a constraint or logical circuit [19, 1], by leveraging compilers from the *knowledge compilation* literature [18, 52]. In a nutshell, $c_K$ is another circuit over variables $S, R, O$ that outputs 1 if an input triple satisfies the encoded logical constraint $K$ and 0 otherwise. See Def. A.2 for a formal definition of such circuits. Then, similarly to what we have showed for computing squared circuits that enable efficient marginalisation (Section 4.2), the satisfaction of compatibility between a GeKC $\phi_{\text{pc}}$ and a constraint circuit $c_K$ enable us to compute $Z_K$ efficiently, as certified by the following theorem.

**Theorem 2** (Tractable integration of constraints in GeKCs). Let $c_K$ be a constraint circuit encoding a logical constraint $K$ over variables $\{S, R, O\}$. Then exactly computing the partition function $Z_K$ of the product $\phi_{\text{pc}}(s, r, o) \cdot c_K(s, r, o) \propto p_K(s, r, o)$ for any GeKC $\phi_{\text{pc}}$ derived from CP, RESCAL, TUCKER or COMPLEX (Section 4) can be done in time $\mathcal{O}((|\mathcal{E}| + |\mathcal{R}|) \cdot |\phi_{\text{pc}}| \cdot |c_K|)$.

In Prop. A.1 we show that the compilation of domain constraints $K$ (Def. 4) is straightforward and results in a constraint circuit $c_K$ having compact size. For example, the size of the constraint circuit encoding the domain constraints of ogbl-biokg is approximately $|c_K| = 307 \cdot 10^3$. To put this number in perspective, the size of the circuit for COMPLEX with embedding size 1000 is the much larger $375 \cdot 10^6$. Since $|c_K|$ is much smaller, by the same argument on the efficiency of GeKCs obtained via squaring (Section 4.3) it results that the integration of logical constraints adds a negligible overhead.

# 6 Related Work

**SOTA KGEs and current limitations.** A plethora of ways to represent and learn KGEs has been proposed, see [31] for a review. KGEs such as CP and COMPLEX are still the de facto go-to choices in many applications [5, 56, 12]. Works performing density estimation in embedding space [78, 11] can sample embeddings, but to sample triples one would need to train a decoder. Several works try to modify training for KGEs as to introduce a penalty for triples that do not satisfy given logical constraints [8, 39, 34, 42, 23, 30], or casting it as a min-max game [44]. Unlike our GeKCs (Section 5), none of these approaches guarantee that test-time predictions satisfy the constraints. Moreover, several heuristics have been proposed to calibrate the probabilistic predictions of KGE models ex-post [61, 79]. As showed in [64], the triple distribution can be modelled autoregressively as $p(S, R, O) = p(S) \cdot p(O \mid S) \cdot p(R \mid S, O)$ where each conditional distribution is encoded by a neural network. However, differently from our GeKCs, integrating constraints exactly or computing *any* marginal (thus conditional) probability is inefficient. KGE models based on non-negative tensor decompositions [63] are equivalent to GeKCs obtained by non-negative restriction (Section 4.1), but are generally trained by minimizing different non-probabilistic losses.

**Circuits.** Circuits provide a unifying framework for several tractable probabilistic models such as sum-product networks (SPNs) and hidden Markov models, which are smooth and decomposable

Table 1: **GeKCs are competitive with their energy-based counterparts.** Best average test MRRs of CP, COMPLEX and GeKCs trained with the PLL and MLE objectives (Eqs. (1) and (2)). For standard deviations and training times see Table F.2.

| Model | FB15k-237 | | WN18RR | | ogbl-biokg | |
|---|---|---|---|---|---|---|
| | PLL | MLE | PLL | MLE | PLL | MLE |
| CP | 0.310 | — | **0.105** | — | 0.831 | — |
| CP$^+$ | 0.237 | 0.230 | 0.027 | 0.026 | 0.496 | 0.501 |
| CP$^2$ | **0.315** | 0.282 | **0.104** | 0.091 | **0.848** | 0.829 |
| COMPLEX | **0.342** | — | **0.471** | — | 0.829 | — |
| COMPLEX$^+$ | 0.214 | 0.205 | 0.030 | 0.029 | 0.503 | 0.516 |
| COMPLEX$^2$ | 0.334 | 0.300 | 0.420 | 0.391 | **0.858** | 0.840 |

PCs [13], as well as compact logical representations [19, 1]. See [75, 13, 17] for an overview. PCs with negative parameters are also called non-monotonic [58], but are surprisingly not as well investigated as their monotonic counterparts, i.e., PCs with only non-negative parameters, at least from the learning perspective. Similarly to our construction for COMPLEX$^+$ (Appendix C.2), Dennis [20] constrains the output of the non-monotonic sub-circuits of a larger PC to be less than their monotonic counterparts. Squaring a circuit has been investigating for tractably computing several divergences [76] and is related to the Born-rule of quantum mechanics [51].

**Circuits for relational data.** Logical circuits to compile formulas in first-order logic (FOL) [25] have been used to reason over relational data, e.g. via exchangeability [68, 49]. Other formalisms such as tractable Markov Logic [77], probabilistic logic bases [50], relational SPNs [46, 45] and generative clausal networks [71] use underlying circuit-like structures to represent probabilistic models over a tractable fragment of FOL formulas. These works assume that every atom in a grounded formula is associated to a random variable, also called the possible world semantics in probabilistic logic programs [57] and databases (PDBs) [14]. In this semantics, TractOR [26] casts answering complex queries over KGEs as to performing inference in PDBs. Differently from these works, our GeKCs are models defined over only three variables (Section 2). In Appendix E we reconcile these two semantics by interpreting the probability of a triple to be proportional to that of all KGs containing it.

## 7 Empirical Evaluation

We aim to answer the following research questions: **RQ1)** are GeKCs competitive with commonly used KGEs for link prediction? **RQ2)** Does integrating domain constraints in GeKCs benefit training and prediction?; **RQ3)** how good are the triples sampled from GeKCs?

### 7.1 Link Prediction (RQ1)

**Experimental setting.** We evaluate GeKCs on standard KG benchmarks for link prediction[4]: FB15k-237 [62], WN18RR [21] and ogbl-biokg [32], whose statistics can be found in Appendix F.1. As usual [48, 56, 54], we assess the models for predicting objects (queries $(s, r, ?)$) and subjects (queries $(?, r, o)$), and report their *mean reciprocal rank* (MRR) and *fraction of hits at $k$* (Hits@$k$) (see Appendix F.2). We remark that our aim in this Section is *not to score the new state-of-the-art link prediction performance on these benchmarks*. Instead, we aim to rigorously assess how close GeKCs can be to commonly used and reasonably tuned KGE models. We focus on CP and COMPLEX as they currently are the go-to models of choice for link prediction [40, 56, 12]. We compare them against our GeKCs CP$^+$, COMPLEX$^+$, CP$^2$ and COMPLEX$^2$ (Section 4). Appendix F.4 collects all the details about the model hyperparameters and training for reproducibility.

**Link prediction results.** Table 1 reports the MRR and times for all benchmarks and models when trained by PLL or MLE. First, CP$^2$ and COMPLEX$^2$ achieve competitive scores when compared to CP and COMPLEX. Moreover, CP$^2$ (resp. COMPLEX$^2$) always outperforms CP$^+$ (resp. COMPLEX$^+$), thus providing empirical evidence that negative embedding values are crucial for model expressiveness. Concerning times, Table F.2 shows that squared GeKCs can train much faster on large KGs (see Section 4.3): CP$^2$ and COMPLEX$^2$ require less than half the training time of CP and COMPLEX on ogbl-biokg, while also unexpectedly scoring the current SOTA MRR on it.[5] We experiment also on the much larger ogbl-wikikg2 KG [32], comprising millions of entities. Even more remarkably, we are

---

[4]Code is available at https://github.com/april-tools/gekcs.

[5]Across non-ensemble methods and according to the OGB leaderboard, updated at the time of this writing.

| Model | $k$ | Embedding size | | | |
|---|---|---|---|---|---|
| | | 10 | 50 | 200 | 1000 |
| COMPLEX | 1 | 99.68 | 99.90 | 99.93 | 99.94 |
| | 20 | 99.81 | 99.79 | 99.85 | 99.91 |
| | 100 | 99.60 | 99.44 | 99.60 | 99.77 |
| COMPLEX$^2$ | 1 | 82.50 | 94.22 | 99.30 | 99.50 |
| | 20 | 86.50 | 96.70 | 99.42 | 99.64 |
| | 100 | 90.66 | 97.71 | 99.23 | 98.78 |
| D-COMPLEX$^2$ | 1 | **100.00** | **100.00** | **100.00** | **100.00** |
| | 20 | **100.00** | **100.00** | **100.00** | **100.00** |
| | 100 | **100.00** | **100.00** | **100.00** | **100.00** |

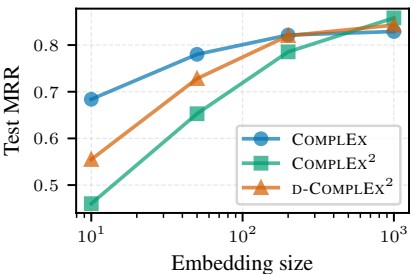

Figure 5: **GeKCs with domain constraints guarantee domain-consistent predictions.** Semantic consistency scores (Sem@$k$) [33] on ogbl-biokg achieved by COMPLEX, COMPLEX$^2$ and its integration with domain constraints (D-COMPLEX$^2$) (left), and MRRs computed on test queries (right). COMPLEX infers 200+ triples violating constraints as the highest scoring completions ($k = 1$).

able to score an MRR of 0.572 after just ~3 hours with COMPLEX$^2$ trained by PLL with a batch size of $10^4$ and embedding size $d = 100$. To put this in context, we were able to score 0.562 with the best configuration of COMPLEX *but after ~3 days*, as we could not fit in memory more than a batch size 500.[6] The same trends are shown for the Hits@$k$ (Table F.3) and likelihood (Table F.4) metrics.

**Distilling GeKCs.** Table F.5 reports the results achieved by CP$^2$ and COMPLEX$^2$ initialised with the parameters of learned CP and COMPLEX (see Section 4.2) and confirms we can quickly turn an EBM into GeKC, thus inheriting all the perks of being a tractable generative model.

**Calibration study.** We also measure how well calibrated the predictions of the models in Table 1 are, which is essential to ensure trustworthiness in critical tasks. For example, given a perfectly calibrated model, for all the triples predicted with a probability of 80%, exactly 80% of them would actually exist [79]. On all KGs but WN18RR, GeKCs achieve lower empirical calibration errors [29] and better calibrated curves than their counterparts, as we report in Appendix F.5.3. The worse performance of all models on WN18RR can be explained by the distribution shift that exists between its training and test split, which we better confirm in Section 7.3.

## 7.2 Integrating Domain Constraints (RQ2)

We focus on ogbl-biokg [32], as it contains the domain metadata for each entity (i.e., disease, drug, function, protein, or side effect). Given the entity domains allowed for each predicate, we formulate domain constraints as in Def. 4. First, we want to estimate how likely are the models to predict triples that do not satisfy the domain constraints. We focus on COMPLEX and COMPLEX$^2$, as they have been shown to achieve the best results in Section 7.1 and introduce D-COMPLEX$^2$ as the constraints-aware version of COMPLEX$^2$ (Section 5). For each test query $(s, r, ?)$ (resp. $(?, r, o)$), we compute the Sem@$k$ score [33] as the average percentage of triples in the first $k$ positions of the rankings of potential object (resp. subject) completions that satisfy the domain constraints (see Appendix F.2).

Fig. 5 highlights how both COMPLEX and COMPLEX$^2$ systematically predict object (or subject) completions that violate domain constraints even for large embedding sizes. For instance, a Sem@1 score of 99% (resp. 99.9%) means that ~3200 (resp. ~320) predicted test triples violate domain constraints. While for COMPLEX and COMPLEX$^2$ there is no theoretical guarantee of consistent predictions with respect to the domain constraints, D-COMPLEX$^2$ always guarantee consistent predictions *by design*. Furthermore, we observe a significant improvement in terms of MRR when integrating constraints for smaller embedding sizes, as reported in Fig. 5.

## 7.3 Quality of sampled triples (RQ3)

Inspired by the literature on evaluating deep generative models for images, we propose a metric akin to the *kernel Inception distance* [4] to evaluate the quality of the triples we can sample with GeKCs.

**Definition 5** (Kernel triple distance (KTD)). Given $\mathbb{P}, \mathbb{Q}$ two probability distributions over triples, and a positive definite kernel $k \colon \mathbb{R}^h \times \mathbb{R}^h \to \mathbb{R}$, we define the *kernel triple distance* KTD$(\mathbb{P}, \mathbb{Q})$ as

---

[6]A smaller ($d = 50$) and highly tuned version of COMPLEX achieves 0.639 MRR but still after days [12].

Table 2: **GeKCs trained by MLE generate new likely triples.** Empirical KTD scores between test triples and triples generated by baselines and GeKCs trained with the PLL objective or by MLE (Eqs. (1) and (2)). Lower is better. For standard deviations see Table F.6.

| Model | FB15k-237 | | WN18RR | | ogbl-biokg | |
|---|---|---|---|---|---|---|
| Training set | 0.055 | | 0.260 | | 0.029 | |
| Uniform | 0.589 | | 0.766 | | 1.822 | |
| NNMFAug | 0.414 | | 0.607 | | 0.518 | |
| | PLL | MLE | PLL | MLE | PLL | MLE |
| CP$^+$ | 0.404 | 0.433 | 0.633 | **0.578** | 0.966 | 0.738 |
| CP$^2$ | 0.253 | **0.070** | 0.768 | 0.768 | 0.039 | **0.017** |
| COMPLEX$^+$ | 0.336 | 0.323 | 0.456 | 0.478 | 0.175 | 0.097 |
| COMPLEX$^2$ | 0.326 | **0.102** | 0.338 | **0.278** | 0.104 | **0.034** |

the squared *maximum mean discrepancy* [28] between triple latent representations obtained via a map $\psi \colon \mathcal{E} \times \mathcal{R} \times \mathcal{E} \to \mathbb{R}^h$ that projects triples to an $h$-dimensional embedding, i.e.,

$$\mathrm{KTD}(\mathbb{P}, \mathbb{Q}) = \mathbb{E}_{x,x' \sim \mathbb{P}}[k(\psi(x), \psi(x'))] + \mathbb{E}_{y,y' \sim \mathbb{Q}}[k(\psi(y), \psi(y'))] - 2 \cdot \mathbb{E}_{x \sim \mathbb{P}, y \sim \mathbb{Q}}[k(\psi(x), \psi(y))].$$

An empirical estimate of the KTD score close to zero indicates that there is little difference between the two triple distributions $\mathbb{P}$ and $\mathbb{Q}$ (see Appendix F.3). For images, $\psi$ is typically chosen as the last embedding of a SOTA neural classifier. We choose $\psi$ to be the $L_2$-normed outputs of the product units of a circuit [73, 74], specifically the SOTA COMPLEX learned by Chen et al. [12] with $h = 4000$. We choose $k$ as the polynomial kernel $k(\mathbf{x}, \mathbf{y}) = (\mathbf{x}^\top \mathbf{y} + 1)^3$, following Binkowski et al. [4].

Table 2 shows the empirical KTD scores computed between the test triples and the generated ones, and Fig. F.1 visualises triple embeddings. We employ two baselines: a uniform probability distribution over all possible triples and NNMFAug [9], the only work to address triple sampling to the best of our knowledge. We also report the KTD scores for training triples as an empirical lower bound. Squared GeKCs achieve lower KTD scores with respect to the ones obtained by non-negative restriction, confirming again a better estimation of the joint distribution. In addition, they achieve far lower KTD scores than all competitors when learning by MLE (Eq. (2)), which justifies its usage as an objective. Lastly, we confirm the distribution shift on WN18RR: training set KTD scores are far from zero, but even in this challenging scenario, COMPLEX$^2$ scores KTD values that are closer to the training ones.

## 8 Conclusions and Future Work

We proposed to re-interpret the representation and learning of widely used KGE models such as CP, RESCAL, TUCKER and COMPLEX, as generative models, overcoming some of the classical limitation of their usual EBM interpretation (see Sections 1 and 2). GeKC-variants for other KGE models whose scores are multilinear maps can be readily devised in the same way. Moreover, we conjecture that other KGE models defining score functions having a distance-based semantics such as TransE [7] and RotatE [60] can be reinterpreted to be GeKCs as well. Our GeKCs open up a number of interesting future directions. First, we plan to investigate how the enhanced efficiency and calibration of GeKCs can help in complex reasoning tasks beyond link prediction [2]. Second, we can leverage the rich literature on learning the structure of circuits [72, 75] to devise smaller and sparser KGE circuit architectures that better capture the triple distribution or sporting structural properties that can make reasoning tasks other than marginalisation efficient [22, 16, 71].

## Acknowledgments and Disclosure of Funding

We would like to acknowledge Iain Murray for his thoughtful feedback and suggestions on a draft of this work. In addition, we thank Xiong Bo and Ricky Zhu for pointing out related works in the knowledge graphs literature. AV was supported by the "UNREAL: Unified Reasoning Layer for Trustworthy ML" project (EP/Y023838/1) selected by the ERC and funded by UKRI EPSRC. RP was supported by the Graz Center for Machine Learning (GraML).

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

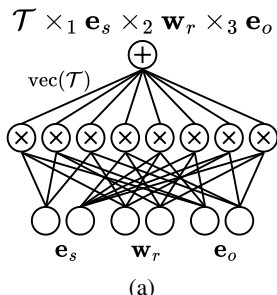
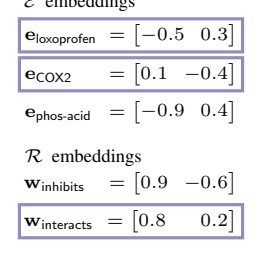
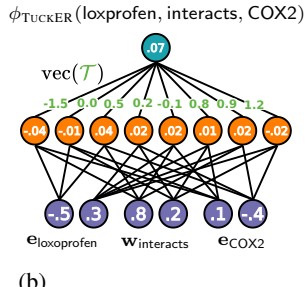

|     |     |
| --- | --- |
| (a) | (b) |

Figure A.1: **Evaluation of circuit representations of score functions as in neural networks.** Feed-forward evaluation of the TUCKER score function as a circuit over 2-dimensional embeddings and parameterised by the core tensor $\mathcal{T}$ (see proof of Prop. 1 below) (a). Given a triple (loxprofen, interacts, COX2), the input units (Def. 1) map subject, predicate and object to their embedding entries (in violet boxes). Then, the circuit is evaluated similarly to neural networks: the products (in orange) are evaluated before the weighted sum (in blue), which is parameterised by the core tensor values (in green) (b). The output of the circuit is the score of the input triple.

## A Proofs

### A.1 KGE Models as Circuits

**Proposition 1** (Score functions of KGE models as circuits). The computational graphs of the score functions $\phi$ of CP, RESCAL, TUCKER and COMPLEX are smooth and decomposable circuits over $\mathbf{X} = \{S, R, O\}$, whose evaluation cost is $\mathrm{cost}(\phi) \in \Theta(|\phi|)$, where $|\phi|$ denotes the number of edges in the circuit, also called its size. For example, the size of the circuit for CP is $|\phi_{\mathrm{CP}}| \in \mathcal{O}(d)$.

*Proof.* We present the proof by construction for TUCKER [3], as CP [40], RESCAL [47] and COMPLEX [65] define score functions that are a specialisation of it [3] (see below). Given a triple $(s, r, o) \in \mathcal{E} \times \mathcal{R} \times \mathcal{E}$, the TUCKER score function computes

$$\phi_{\mathrm{TUCKER}}(s, r, o) = \mathcal{T} \times_1 \mathbf{e}_s \times_2 \mathbf{w}_r \times_3 \mathbf{e}_o = \sum_{i=1}^{d_e} \sum_{j=1}^{d_r} \sum_{k=1}^{d_e} \tau_{ijk} e_{si} w_{rj} e_{ok} \qquad (4)$$

where $\mathcal{T} \in \mathbb{R}^{d_e \times d_r \times d_e}$ denotes the core tensor, $\times_n$ denotes the tensor product along the $n$-th mode, and $d_e, d_r$ denote respectively the embedding sizes of entities and predicates (which might not be equal). To see how this parametrization generalises that of CP, RESCAL and COMPLEX, consider for example the score function of CP on $d$-dimensional embeddings. It can be obtained by (i) setting the core tensor $\mathcal{T}$ to be a *diagonal tensor* having ones on the superdiagonal, and (ii) having two distinct embedding instances for each entity that are used depending on their role (either subject or object) in a triple. The embeddings $\mathbf{e}_s, \mathbf{e}_o \in \mathbb{R}^{d_e}$ (resp. $\mathbf{w}_r \in \mathbb{R}^{d_r}$) are rows of the matrix $\mathbf{E} \in \mathbb{R}^{|\mathcal{E}| \times d_e}$ (resp. $\mathbf{W} \in \mathbb{R}^{|\mathcal{R}| \times d_r}$), which associates an embedding to each entity (resp. predicate).

**Constructing the circuit.** For the construction of the equivalent circuit it suffices to (i) create an input unit for each $i$-th entry of an embedding for subjects, predicates and objects, as to implement a look-up table that computes the corresponding embedding value for an entity or predicate, and (ii) transform the tensor multiplications into corresponding sum and product units. We start by introducing the input units $l_i^S$, $l_j^R$ and $l_k^O$ for $1 \le i, k \le d_e$ and $1 \le j \le d_r$ as parametric mappers over variables $S$, $R$ and $O$, respectively. The input units $l_i^S$ and $l_k^O$ (resp. $l_j^R$) are parameterised by the matrix $\mathbf{E}$ (resp. $\mathbf{W}$) such that $l_i^S(s; \mathbf{E}) = e_{si}$ and $l_i^O(o; \mathbf{E}) = e_{oi}$ for some $s, o \in \mathcal{E}$ (resp. $l_j^R(r; \mathbf{W}) = w_{ri}$ for some $r \in \mathcal{R}$). To encode the tensor products in Eq. (4) we introduce $d_e^2 \cdot d_r$ product units $\phi_{ijk}$, each of them computing the product of a combination of the outputs of input units.

$$\phi_{ijk}(s, r, o) = l_i^S(s) \cdot l_j^R(r) \cdot l_k^O(o)$$

Finally, a sum unit $\phi_{\mathrm{out}}$ parameterised by the core tensor $\mathcal{T} \in \mathbb{R}^{d_e \times d_r \times d_r}$ computes a weighted summation of the outputs given by the product units, i.e.,

$$\phi_{\mathrm{out}}(s, r, o) = \sum_{\substack{(i,j,k) \in \\ [d_e] \times [d_r] \times [d_e]}} \tau_{ijk} \cdot \phi_{ijk}(s, r, o)$$

Table A.1: **Score functions as compact circuits.** Asymptotic size of circuits encoding the score functions of CP, RESCAL, COMPLEX and TUCKER, with respect to the embedding size. For TUCKER, $d_e$ and $d_r$ denote the embedding sizes for entities and predicates, respectively.

| KGE Model | Circuit Size | KGE Model | Circuit Size |
|-----------|--------------|-----------|--------------|
| CP | $\mathcal{O}(d)$ | RESCAL | $\mathcal{O}(d^2)$ |
| COMPLEX | $\mathcal{O}(d)$ | TUCKER | $\mathcal{O}(d_e^2 \cdot d_r)$ |

where $[d]$ denotes the set $\{1, \ldots, d\}$ and $\tau_{ijk}$ is the $(i, j, k)$-th entry of $\mathcal{T}$. We now observe that the constructed circuit $\phi_{\text{out}}$ encodes the TUCKER score function (Eq. (4)), as $\phi_{\text{TUCKER}}(s, r, o) = \phi_{\text{out}}(s, r, o)$ for any input triple $(s, r, o) \in \mathcal{E} \times \mathcal{R} \times \mathcal{E}$.

**Circuit evaluation and properties.** Evaluating the score function of TUCKER corresponds to performing a feed-forward pass of its circuit representation, where each computational unit is evaluated once, as we illustrate in Fig. A.1. As such, the cost of evaluating the score function is proportional to the size of its circuit representation, i.e., $\text{cost}(\phi) \in \Theta(|\phi|)$ where $|\phi| \in \mathcal{O}(d_e^2 \cdot d_r)$ is the number of edges. In Table A.1 we show how the sizes of the circuit representation of the other score functions increases with respect to the embedding size. Finally, since each product unit $\phi_{ijk}$ is defined on the same scope (see Def. 1) $\{S, R, O\}$ and fully decompose it into its inputs (i.e., into $\{S\}, \{R\}, \{O\}$), and the inputs of the sum unit $\phi_{\text{out}}$ are all defined over the same scope, we have that the circuit satisfies smoothness and decomposability (Def. 2). $\qquad\square$

Furthermore, in Lem. A.1 we show that the circuit representations of CP, RESCAL, TUCKER and COMPLEX and the proposed GeKCs (Section 4) satisfy a structural property known as *omni-compatibility* (see Def. B.2). In a nutshell, the score functions of the aforementioned KGE models and GeKCs are circuits that fully decompose their scope $\{S, R, O\}$ into $(\{S\}, \{R\}, \{O\})$. The satisfaction of this property will be useful to prove both Thm. 1 and Thm. 2 later in this appendix.

**Lemma A.1** (KGE models and derived GeKCs are omni-compatible)**.** The circuit representation of the score functions of CP, RESCAL, TUCKER, COMPLEX and their GeKCs counterparts obtained by non-negative restriction (Section 4.1) or squaring (Section 4.2) are omni-compatible (see Def. B.2).

*Proof.* To begin, we note that to comply with Def. B.2 every omni-compatible circuit shall contain product units that fully factorise over their scope. In other words, for every product unit $n$ with scope $\text{sc}(n) = \mathbf{X}$, its scope shall decompose as $(\{X_1\}, \{X_2\}, \ldots, \{X_{|\text{sc}(n)|}\})$. To see why, consider a circuit $\phi$ with a product unit $n$ whose scope is decomposed as $\text{sc}(n) = (\mathbf{X}, \mathbf{Y})$. It is easy to construct another circuit $\phi'$ that is not compatible with $\phi$ by having a product unit $m$ with scope $\text{sc}(m) = \text{sc}(n)$ decomposed in a way that it cannot be rearranged by introducing additional decomposable product units (Def. 2), e.g., $\text{sc}(m) = (\mathbf{Z}, \mathbf{W})$ with $\mathbf{Z} \cap \mathbf{X} \neq \varnothing$ and $\mathbf{W} \cap \mathbf{X} \neq \varnothing$. As such, every omni-compatible circuit over $\mathbf{X}$ must be representable in the form $\sum_{i=1}^{N} \theta_i \prod_{k=1}^{|\mathbf{X}|} l_{ik}(X_k)$ without any increase in its size.

Now, it is easy to verify that the circuit representations of CP, RESCAL, TUCKER and COMPLEX follow the above form, with a different number $N$ of product units feeding the single sum unit, but each one decomposing its scope $\{S, R, O\}$ into $(\{S\}, \{R\}, \{O\})$ (see Fig. 2). From this it follows that CP⁺, RESCAL⁺, TUCKER⁺ and COMPLEX⁺ are omni-compatible as well, as they share the same structure of their energy-based counterpart, while just enforcing non-negative activations via reparametrisation (see Section 4.1).

Finally, we note that CP², RESCAL², TUCKER² and COMPLEX² are still omni-compatible because the square operation yields the following fully-factorised representation: $(\sum_{i=1}^{N} \theta_i \prod_{k=1}^{|\mathbf{X}|} l_{ik}(X_k))^2 = \sum_{i=1}^{N} \sum_{j=1}^{N} \theta_i \theta_j \prod_{k=1}^{|\mathbf{X}|} l_{ik}(X_k) \prod_{k=1}^{|\mathbf{X}|} l_{jk}(X_k)$ which can be easily rewritten as $\sum_{h=1}^{N^2} \omega_h \prod_{k=1}^{|\mathbf{X}|} l_{hk}(X_k)$ where now $h$ ranges over the Cartesian product of $i \in [N]$ and $j \in [N]$, $\omega_h$ is the product of $\theta_i \theta_j$ and $l_{hk}$ is a new input unit that encodes $l_{ik}(X_k) l_{jk}(X_k)$ for a certain variable index $k$. $\qquad\square$

## A.2 Efficient Summations over Circuits

**Proposition 2** (Efficient Summations). Let $\phi$ be a smooth and decomposable circuit over $\mathbf{X} = \{S, R, O\}$ that encodes the score function of a KGE model. The sum $\sum_{s \in \mathcal{E}} \sum_{r \in \mathcal{R}} \sum_{o \in \mathcal{E}} \phi(s, r, o)$ or any other summation over subjects, predicates or objects can be computed in time $\mathcal{O}((|\mathcal{E}| + |\mathcal{R}|) \cdot |\phi|)$.

*Proof.* A proof for the computation of marginal probabilities in smooth and decomposable *probabilistic circuits* (PCs) defined over discrete variables in linear time with respect to their size can be found in [13]. This proof also applies for computing summations in smooth and decomposable circuits that do not necessarily corresponds to marginal probabilities [76]. The satisfaction of smoothness and decomposability (Def. 2) in a circuit $\phi$ permits to push outer summations inside the computational graph until input units are reached, where summations are actually performed independently and on smaller sets of variables (i.e., $\{S\}, \{R\}, \{O\}$ in our case), and then to evaluate the circuit only once.

Here we take into account the computational cost of summing over each input unit (see proof of Prop. 1), which is $\mathcal{O}(|\mathcal{E}|)$ (resp. $\mathcal{O}(|\mathcal{R}|)$) for those defined on variables $S, O$ (resp. $R$). Since the size of the circuit $|\phi|$ must be at least the number of input units, we retrieve that the overall complexity for computing summations as stated in the proposition is $\mathcal{O}((|\mathcal{E}| + |\mathcal{R}|) \cdot |\phi|)$.

As an example, consider the CP score function computing $\phi_{\mathrm{CP}}(s, r, o) = \langle \mathbf{e}_s, \mathbf{w}_r, \mathbf{e}_o \rangle$ for some triple $(s, r, o)$ and embeddings $\mathbf{e}_s, \mathbf{w}_r, \mathbf{e}_o \in \mathbb{R}^d$. We can compute $\sum_{s \in \mathcal{E}} \sum_{r \in \mathcal{R}} \sum_{o \in \mathcal{E}} \phi_{\mathrm{CP}}(s, r, o)$ by pushing the outer summations inside the trilinear product, i.e., by computing it as $\langle \sum_{s \in \mathcal{E}} \mathbf{e}_s, \sum_{r \in \mathcal{R}} \mathbf{w}_r, \sum_{o \in \mathcal{E}} \mathbf{e}_o \rangle$, which requires time $\mathcal{O}((|\mathcal{E}| + |\mathcal{R}|) \cdot d)$. $\qquad \square$

## A.3 Efficient Summations over Squared Circuits

**Theorem 1** (Efficient summations of squared GeKCs). Performing summations as stated in Prop. 2 on $\mathrm{CP}^2$, $\mathrm{RESCAL}^2$, $\mathrm{TUCKER}^2$ and $\mathrm{COMPLEX}^2$ can be done in time $\mathcal{O}((|\mathcal{E}| + |\mathcal{R}|) \cdot |\phi|^2)$.

*Proof.* In Lem. A.1 we showed that the circuit representations $\phi$ of CP, RESCAL, TUCKER and COMPLEX are omni-compatible (see Def. B.2). As a consequence, $\phi$ is compatible (see Def. B.1) with itself. Therefore, Prop. B.1 ensures that we can construct the product circuit $\phi \cdot \phi$ (i.e., $\phi^2$) as a smooth and decomposable circuit having size $\mathcal{O}(|\phi|^2)$ in time $\mathcal{O}(|\phi|^2)$. Since $\phi^2$ is still smooth and decomposable, Prop. 2 guarantees that we can perform summations in time $\mathcal{O}((|\mathcal{E}| + |\mathcal{R}|) \cdot |\phi|^2)$. $\quad \square$

## A.4 Circuits encoding Domain Constraints

In Def. A.1 we introduce the concepts of *support* and *determinism*, whose definition is useful to describe *constraint circuits* in Def. A.2.

**Definition A.1** (Support and Determinism [13, 76]). In a circuit the *support* of a computational unit $n$ over variables $\mathbf{X}$ computing $\phi_n(\mathbf{X})$ is defined as the set of value assignments to variables in $\mathbf{X}$ such that the output of $n$ is non-zero, i.e., $\mathrm{supp}(n) = \{\mathbf{x} \in \mathrm{val}(\mathbf{X}) \mid \phi_n(\mathbf{X}) \neq 0\}$. A sum unit $n$ is *deterministic* if its inputs have disjoint *supports*, i.e., $\forall i, j \in \mathrm{in}(n), i \neq j : \mathrm{supp}(i) \cap \mathrm{supp}(j) = \varnothing$.

**Definition A.2** (Constraint Circuit [1]). Given a propositional logic formula $K$, a *constraint circuit* $c_K$ is a smooth and decomposable PC over variables $\mathbf{X}$ with *deterministic* sum units (Def. A.1) and indicator functions as input units, such that $c_K(\mathbf{x}) = \mathbb{1}\{\mathbf{x} \models K\}$ for any $\mathbf{x} \in \mathrm{val}(\mathbf{X})$.

In general, we can compile any propositional logic formula into a constraint circuit (Def. A.2) by leveraging knowledge compilation techniques [19, 18, 52]. For domain constraints (Def. 4) this compilation process is straightforward, as we detail in the following proposition and proof.

**Proposition A.1** (Circuit encoding domain constraints). Let $K = K_{r_1} \vee \ldots \vee K_{r_m}$ be a disjunction of domain constraints defined over a set of predicates $\mathcal{R} = \{r_1, \ldots, r_m\}$ and a set of entities $\mathcal{E}$ (Def. 4). We can compile $K$ into a constraint circuit $c_K$ (Def. A.2) defined over variables $\mathbf{X} = \{S, R, O\}$ having size $\mathcal{O}(|\mathcal{E}| \cdot |\mathcal{R}|)$ in the worst case and $\mathcal{O}(|\mathcal{E}| + |\mathcal{R}|)$ in the best case.

*Proof.* Let $K = K_{r_1} \vee \ldots \vee K_{r_m}$ be a disjunction of domain constraints (Def. 4) where

$$K_r \equiv S \in \kappa_S(r) \wedge R = r \wedge O \in \kappa_O(r) \equiv (\vee_{u \in \kappa_S(r)} S = u) \wedge R = r \wedge (\vee_{v \in \kappa_O(r)} O = v).$$

Note that the disjunctions in $K$ are deterministic, i.e., only one of their argument can be true at the same time. This enables us to construct the constraint circuit $c_K$ such that $c_K(s, r, o) = \mathbb{1}\{(s, r, o) \models K\}$ for any triple by simply replacing conjunctions and disjunctions with product and sum units, respectively. Note that $c_K$ is indeed smooth and decomposable (Def. 2), as the inputs of the sum units are product units having scope $\{S, R, O\}$ that are fully factorised into $(\{S\}, \{R\}, \{O\})$. Moreover, $K$ is a disjunction of $|\mathcal{R}|$ conjunctive formulae having $\mathcal{O}(|\mathcal{E}|)$ terms, and therefore $|c_K| = \mathcal{O}(|\mathcal{E}| \cdot |\mathcal{R}|)$ in the worst case. In the best case of every predicate sharing the same subject and object domains $\kappa_S, \kappa_O \subseteq \mathcal{E}$, we can simplify $K$ into a conjunction of three disjunctive expressions, i.e.,

$$K \equiv (\vee_{u \in \kappa_S} S = u) \wedge (\vee_{r \in \mathcal{R}} R = r) \wedge (\vee_{v \in \kappa_O} O = v)$$

that can be easily compiled into a constraint circuit $c_K$ having size $\mathcal{O}(|\mathcal{E}| + |\mathcal{R}|)$, by again noticing that disjunctions are deterministic. In real-world KGs like ogbl-biokg [32] several predicates share the same subject and object domains, and this permits to have much smaller constraint circuits. $\qquad\square$

## A.5 Efficient Integration of Domain Knowledge in GeKCs

**Theorem 2** (Tractable integration of constraints in GeKCs)**.** Let $c_K$ be a constraint circuit encoding a logical constraint $K$ over variables $\{S, R, O\}$. Then exactly computing the partition function $Z_K$ of the product $\phi_{\mathsf{pc}}(s, r, o) \cdot c_K(s, r, o) \propto p_K(s, r, o)$ for any GeKC $\phi_{\mathsf{pc}}$ derived from CP, RESCAL, TUCKER or COMPLEX (Section 4) can be done in time $\mathcal{O}((|\mathcal{E}| + |\mathcal{R}|) \cdot |\phi_{\mathsf{pc}}| \cdot |c_K|)$.

*Proof.* In Lem. A.1 we showed that the GeKCs $\phi_{\mathsf{pc}}$ derived from CP, RESCAL, TUCKER and COMPLEX via non-negative restriction (Section 4.1) or squaring (Section 4.2) are omni-compatible (see Def. B.2). As a consequence, $\phi_{\mathsf{pc}}$ is always compatible with $c_K$ regardless of the encoded logical constraint $K$, since constraint circuits are by definition smooth and decomposable (Def. A.2). By applying Prop. B.1, we retrieve that we can construct $\phi_{\mathsf{pc}} \cdot c_K$ as a smooth and decomposable circuit of size $\mathcal{O}(|\phi_{\mathsf{pc}}| \cdot |c_K|)$ and in time $\mathcal{O}(|\phi_{\mathsf{pc}}| \cdot |c_K|)$. As the resulting product circuit is smooth and decomposable, Prop. 2 guarantees that we can compute its partition function $Z_K = \sum_{s \in \mathcal{E}} \sum_{r \in \mathcal{R}} \sum_{o \in \mathcal{E}} (\phi_{\mathsf{pc}}(s, r, o) \cdot c_K(s, r, o))$ in time $\mathcal{O}((|\mathcal{E}| + |\mathcal{R}|) \cdot |\phi_{\mathsf{pc}}| \cdot |c_K|)$. $\qquad\square$

# B Circuits

## B.1 Tractable Product of Circuits

In this section, we provide the formal definition of *compatibility* (Def. B.1) and *omni-compatibility* (Def. B.2), as stated by Vergari et al. [76]. Given two compatible circuits, Prop. B.1 guarantees that we can represent their product as a smooth and decomposable circuit efficiently.

**Definition B.1** (Compatibility)**.** Two circuits $\phi, \phi'$ over variables $\mathbf{X}$ are *compatible* if (1) they are smooth and decomposable, and (2) any pair of product units $n \in \phi, m \in \phi'$ having the same scope can be rearranged into binary products that are mutually compatible and decompose their scope in the same way, i.e., $(\mathsf{sc}(n) = \mathsf{sc}(m)) \implies (\mathsf{sc}(n_i) = \mathsf{sc}(m_i), n_i$ and $m_i$ are compatible) for some rearrangements of the inputs of $n$ (resp. $m$) into $n_1, n_2$ (resp. $m_1, m_2$).

**Definition B.2** (Omni-compatibility)**.** A circuit $\phi$ over variables $\mathbf{X}$ is *omni-compatible* if it is compatible with any smooth and decomposable circuit over $\mathbf{X}$.

**Proposition B.1** (Tractable product of circuits)**.** Let $\phi, \phi'$ be two compatible (Def. B.1) circuits. We can represent the product circuit $\phi \cdot \phi'$ computing the product of the outputs of $\phi$ and $\phi'$ as a smooth and decomposable circuit having size $\mathcal{O}(|\phi| \cdot |\phi'|)$ in time $\mathcal{O}(|\phi| \cdot |\phi'|)$. Moreover, if both $\phi$ and $\phi'$ are omni-compatible (Def. B.2), then also the product circuit $\phi \cdot \phi'$ is omni-compatible.

Prop. B.1 allows us to compute the partition function and any other marginal probability in GeKCs obtained via squaring efficiently (see Section 4.2 and Thm. 1). In addition, Prop. B.1 is a crucial theoretical result that allows us to inject logical constraints in GeKCs in a way that enable computing the partition function exactly and efficiently (see Section 5 and Thm. 2).

## C  From KGE Models to PCs

### C.1  Interpreting Non-negative Embedding Values

In Fig. C.1 we interpret the embedding values of GeKCs obtained via non-negative restriction – CP⁺, RESCAL⁺, TUCKER⁺, COMPLEX⁺– (Section 4.1) as the parameters of unnormalised categorical distributions over entities (elements in $\mathcal{E}$) or predicates (elements in $\mathcal{R}$).

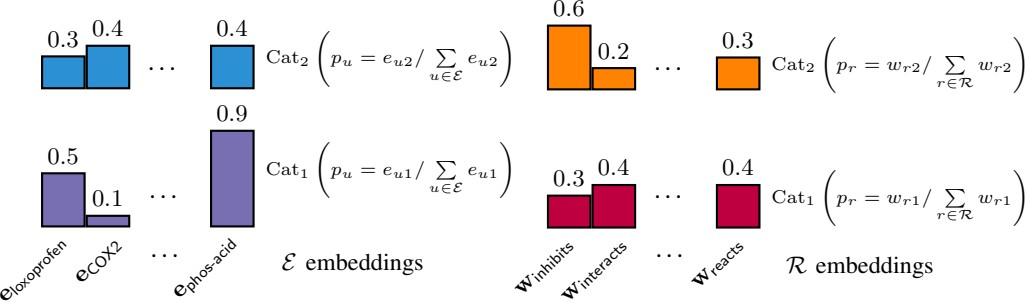

Figure C.1: **Non-negative embeddings parameterise categorical distributions.** 2-dimensional embeddings of GeKCs obtained via non-negative restrictions (Section 4.1) can be seen as the parameters of two categorical distributions over entities (left) or predicates (right) up to renormalisation.

### C.2  Realising the Non-negative Restriction of COMPLEX

As anticipated in Section 4.1, for the COMPLEX [65] score function restricting the real and imaginary parts to be non-negative is not sufficient to obtain a PC due to the presence of a subtraction, as showed in the following equation.

$$\phi_{\text{COMPLEX}}(s, r, o) = \langle \text{Re}(\mathbf{e}_s), \text{Re}(\mathbf{w}_r), \text{Re}(\mathbf{e}_o) \rangle + \langle \text{Im}(\mathbf{e}_s), \text{Re}(\mathbf{w}_r), \text{Im}(\mathbf{e}_o) \rangle \\ + \langle \text{Re}(\mathbf{e}_s), \text{Im}(\mathbf{w}_r), \text{Im}(\mathbf{e}_o) \rangle - \langle \text{Im}(\mathbf{e}_s), \text{Im}(\mathbf{w}_r), \text{Re}(\mathbf{e}_o) \rangle \tag{5}$$

Here $\mathbf{e}_s, \mathbf{w}_r, \mathbf{e}_o \in \mathbb{C}^d$ are the embeddings associated to the subject, predicate and object, respectively. Under the restriction of embedding values to be non-negative, we ensure that $\phi_{\text{COMPLEX}}(s, r, o) \geq 0$ for any input triple by enforcing the additional constraint

$$\langle \text{Re}(\mathbf{e}_s), \text{Re}(\mathbf{w}_r), \text{Re}(\mathbf{e}_o) \rangle \geq \langle \text{Im}(\mathbf{e}_s), \text{Im}(\mathbf{w}_r), \text{Re}(\mathbf{e}_o) \rangle, \tag{6}$$

which can be simplified into the two distinct inequalities

$$\forall u \in \mathcal{E} \quad \text{Re}(e_{ui}) \geq \text{Im}(e_{ui}) \qquad \text{and} \qquad \forall r \in \mathcal{R} \quad \text{Re}(w_{ri}) \geq \text{Im}(w_{ri}).$$

In other words, we want the real part of each embedding value to be always greater or equal than the corresponding imaginary part. We implement this constraint in practice by reparametrisation of the imaginary part in function of the real part, i.e.,

$$\forall u \in \mathcal{E} \quad \text{Im}(e_{ui}) = \text{Re}(e_{ui}) \cdot \sigma(\theta_{ui}) \tag{7}$$
$$\forall r \in \mathcal{R} \quad \text{Im}(w_{ri}) = \text{Re}(w_{ri}) \cdot \sigma(\gamma_{ri}) \tag{8}$$

where $\sigma(x) = 1/(1 + \exp(-x)) \in [0, 1]$ denotes the logistic function and $\theta_{ui}, \gamma_{ri} \in \mathbb{R}$ are additional parameters associated to entities $u \in \mathcal{E}$ and predicates $r \in \mathcal{R}$, respectively. The reparametrisation of the imaginary parts using Eqs. (7) and (8) is a sufficient condition for the satisfaction of the constraint showed in Eq. (6), and also maintains the same number of learnable parameters of COMPLEX.

### C.3  Sampling from GeKCs with Non-negative Parameters

**Parameters interpretation.**    Sum units with non-negative parameters in smooth and decomposable PCs can be seen as marginalised discrete latent variables, similarly to the latent variable interpretation in mixture models [55, 53]. That is, the non-negative parameters of a sum unit are the parameters of a (possibly unnormalised) categorical distribution over assignments to a latent variable. For CP⁺ and RESCAL⁺ (Section 4.1), the non-negative parameters of the sum unit encode a uniform and

unnormalised categorical distribution, as they are all fixed to 1 (see Fig. 2). By contrast, in TUCKER$^+$ these parameters are the vectorisation of the core tensor $\mathcal{T}$ (see the proof of Prop. 1), and hence they are learned. The input units of CP$^+$, RESCAL$^+$ and TUCKER$^+$ can be interpreted as unnormalised categorical distribution over entities or predicates, as detailed in Appendix C.1.

**Sampling from CP$^+$, COMPLEX$^+$, TUCKER$^+$.** Thanks to the latent variable interpretation, ancestral sampling in CP$^+$, RESCAL$^+$ and TUCKER$^+$ can be performed by (1) sampling an assignment to the latent variable associated to the single sum unit, i.e., one of its input branches, (2) selecting the corresponding combination of subject-predicate-object input units, and (3) sampling a subject, predicate and object respectively from each of the indexed unnormalized categorical distributions.

## C.4 Learning Complexity of GeKCs

In Table C.1 we summarise the complexities of computing the PLL and MLE objectives (Eqs. (1) and (2)) for KGE models and GeKCs. Asymptotically, GeKCs manifest better time and space complexities with respect to the number of entities $|\mathcal{E}|$, batch size $|B|$ and embedding size. This makes GeKCs more efficient than traditional KGE models during training, both in time and memory (see Section 4.3 and Fig. 3).

Table C.1: **Summary of complexities for exactly computing the PLL and MLE objectives.** Time and space complexity of computing $\log p(o \mid s, r)$ and the partition function $Z$. These complexities are respectively lower bounds of the complexities of computing the PLL and MLE objectives, as we have that $|\mathcal{E}| \gg |\mathcal{R}|$ for large real-world KGs. For CP, RESCAL and COMPLEX and GeKCs derived from them, $d$ denotes the size of both entity and predicate embeddings. For TUCKER and GeKCs derived from it, $d_e$ and $d_r$ denote the embedding sizes for entities and predicates, respectively.

| Model | Complexity of $\log p(o \mid s, r)$ | | Complexity of $Z$ | |
|---|---|---|---|---|
| | **Time** | **Space** | **Time** | **Space** |
| CP | $\mathcal{O}(\|\mathcal{E}\| \cdot \|B\| \cdot d)$ | $\mathcal{O}(\|\mathcal{E}\| \cdot \|B\|)$ | $\mathcal{O}(\|\mathcal{E}\|^2 \cdot \|\mathcal{R}\| \cdot d)$ | $\mathcal{O}(d)$ |
| RESCAL | $\mathcal{O}(\|\mathcal{E}\| \cdot \|B\| \cdot d + \|B\| \cdot d^2)$ | $\mathcal{O}(\|\mathcal{E}\| \cdot \|B\|)$ | $\mathcal{O}(\|\mathcal{E}\|^2 \cdot \|\mathcal{R}\| \cdot d^2)$ | $\mathcal{O}(d^2)$ |
| TUCKER | $\mathcal{O}(\|\mathcal{E}\| \cdot \|B\| \cdot d_e + \|B\| \cdot d_e^2 \cdot d_r)$ | $\mathcal{O}(\|\mathcal{E}\| \cdot \|B\|)$ | $\mathcal{O}(\|\mathcal{E}\|^2 \cdot \|\mathcal{R}\| \cdot d_e^2 \cdot d_r)$ | $\mathcal{O}(d_e^2 \cdot d_r)$ |
| COMPLEX | $\mathcal{O}(\|\mathcal{E}\| \cdot \|B\| \cdot d)$ | $\mathcal{O}(\|\mathcal{E}\| \cdot \|B\|)$ | $\mathcal{O}(\|\mathcal{E}\|^2 \cdot \|\mathcal{R}\| \cdot d)$ | $\mathcal{O}(d)$ |
| CP$^+$ | $\mathcal{O}((\|\mathcal{E}\| + \|B\|) \cdot d)$ | $\mathcal{O}(\|B\| \cdot d)$ | $\mathcal{O}((\|\mathcal{E}\| + \|\mathcal{R}\|) \cdot d)$ | $\mathcal{O}(d)$ |
| RESCAL$^+$ | $\mathcal{O}((\|\mathcal{E}\| + \|B\| \cdot d) \cdot d)$ | $\mathcal{O}(\|B\| \cdot d^2)$ | $\mathcal{O}((\|\mathcal{E}\| + \|\mathcal{R}\| \cdot d) \cdot d)$ | $\mathcal{O}(d^2)$ |
| TUCKER$^+$ | $\mathcal{O}((\|\mathcal{E}\| + \|B\| \cdot d_e \cdot d_r) \cdot d_e)$ | $\mathcal{O}(\|B\| \cdot d_e \cdot d_r)$ | $\mathcal{O}(\|\mathcal{E}\| \cdot d_e + \|\mathcal{R}\| \cdot d_r + d_e^2 \cdot d_r)$ | $\mathcal{O}(d_e^2 \cdot d_r)$ |
| COMPLEX$^+$ | $\mathcal{O}((\|\mathcal{E}\| + \|B\|) \cdot d)$ | $\mathcal{O}(\|B\| \cdot d)$ | $\mathcal{O}((\|\mathcal{E}\| + \|\mathcal{R}\|) \cdot d)$ | $\mathcal{O}(d)$ |
| CP$^2$ | $\mathcal{O}((\|\mathcal{E}\| + \|B\|) \cdot d^2)$ | $\mathcal{O}(\|B\| \cdot d)$ | $\mathcal{O}((\|\mathcal{E}\| + \|\mathcal{R}\|) \cdot d^2)$ | $\mathcal{O}(d^2)$ |
| RESCAL$^2$ | $\mathcal{O}((\|\mathcal{E}\| + \|B\|) \cdot d^2)$ | $\mathcal{O}(\|B\| \cdot d^2)$ | $\mathcal{O}((\|\mathcal{E}\| + \|\mathcal{R}\| \cdot d) \cdot d^2)$ | $\mathcal{O}(\|\mathcal{R}\| \cdot d^2)$ |
| TUCKER$^2$ | $\mathcal{O}((\|\mathcal{E}\| + \|B\| \cdot d_r) \cdot d_e^2)$ | $\mathcal{O}(\|B\| \cdot d_e \cdot d_r)$ | $\mathcal{O}(\|\mathcal{E}\| \cdot d_e^2 + \|\mathcal{R}\| \cdot d_r^2 + d_e^2 \cdot d_r)$ | $\mathcal{O}(d_e^2 \cdot d_r)$ |
| COMPLEX$^2$ | $\mathcal{O}((\|\mathcal{E}\| + \|B\|) \cdot d^2)$ | $\mathcal{O}(\|B\| \cdot d)$ | $\mathcal{O}((\|\mathcal{E}\| + \|\mathcal{R}\|) \cdot d^2)$ | $\mathcal{O}(d^2)$ |

### C.4.1 Computing the Partition Function

In this section we derive the computational complexity of computing the partition function for GeKCs obtained via squaring (Section 4.2). For a summary of these complexities, see Table C.1.

**CP$^2$ and COMPLEX$^2$.** Here we derive the partition function of CP$^2$. For COMPLEX$^2$ the derivation is similar, as the score function of COMPLEX can be written in terms of trilinear products just like CP (see Eq. (5)). The score function $\phi_{\text{CP}^2}$ encoded by CP$^2$ can be written as

$$\phi_{\text{CP}^2}(s, r, o) = \langle \mathbf{e}_s, \mathbf{w}_r, \mathbf{e}_o \rangle^2 = \sum_{i=1}^{d} \sum_{j=1}^{d} e_{si} e_{sj} w_{ri} w_{rj} e_{oi} e_{oj}$$

where $\mathbf{e}_s, \mathbf{e}_o \in \mathbb{R}^d$ (resp. $\mathbf{w}_r \in \mathbb{R}^d$) are rows of the matrices $\mathbf{U}, \mathbf{V} \in \mathbb{R}^{|\mathcal{E}| \times d}$ (resp. $\mathbf{W} \in \mathbb{R}^{|\mathcal{R}| \times d}$), which associate to each entity (resp. predicate) a vector. By leveraging the *einsum notation* for

brevity, the partition function of $\phi_{\mathrm{CP}^2}$ can be written as

$$Z = \sum_{s \in \mathcal{E}} \sum_{r \in \mathcal{R}} \sum_{o \in \mathcal{E}} \phi_{\mathrm{CP}^2}(s, r, o) = \sum_{i=1}^{d} \sum_{j=1}^{d} \left( \sum_{s \in \mathcal{E}} e_{si} e_{sj} \right) \left( \sum_{r \in \mathcal{R}} w_{ri} w_{rj} \right) \left( \sum_{o \in \mathcal{E}} e_{oi} e_{oj} \right)$$
$$= \mathbf{U}'_{ij} \mathbf{W}'_{ij} \mathbf{V}'_{ij}$$

where $\mathbf{U}' = \mathbf{U}^\top \mathbf{U}$, $\mathbf{W}' = \mathbf{W}^\top \mathbf{W}$ and $\mathbf{V}' = \mathbf{V}^\top \mathbf{V}$ are $d \times d$ matrices. With the simplest algorithm for matrix multiplication, we recover that computing $Z$ requires time $\mathcal{O}(|\mathcal{E}| \cdot d^2 + |\mathcal{R}| \cdot d^2)$ and additional space $\mathcal{O}(d^2)$.

**RESCAL².** The score function $\phi_{\mathrm{RESCAL}^2}$ encoded by RESCAL$^2$ can be written as

$$\phi_{\mathrm{RESCAL}^2}(s, r, o) = \left( \mathbf{e}_s^\top \mathbf{W}_r \mathbf{e}_o \right)^2 = \sum_{(i,j,k,l) \in [d]^4} e_{si} e_{sk} w_{rij} w_{rkl} e_{oj} e_{ol}$$

where $[d]$ denotes the set $\{1, \ldots, d\}$, $\mathbf{e}_s, \mathbf{e}_o \in \mathbb{R}^d$ are rows of the matrix $\mathbf{E} \in \mathbb{R}^{|\mathcal{E}| \times d}$ and $\mathbf{W}_r \in \mathbb{R}^{d \times d}$ are slices along the first mode of the tensor $\mathcal{W} \in \mathbb{R}^{|\mathcal{R}| \times d \times d}$, which consists of stacked matrix embeddings associated to predicates. The partition function of $\phi_{\mathrm{RESCAL}^2}$ can be written as

$$Z = \sum_{s \in \mathcal{E}} \sum_{r \in \mathcal{R}} \sum_{o \in \mathcal{E}} \phi_{\mathrm{RESCAL}^2}(s, r, o) = \sum_{(i,j,k,l) \in [d]^4} \left( \sum_{s \in \mathcal{E}} e_{si} e_{sk} \right) \left( \sum_{r \in \mathcal{R}} w_{rij} w_{rkl} \right) \left( \sum_{o \in \mathcal{E}} e_{oj} e_{ol} \right)$$
$$= \mathbf{E}'_{ik} \mathcal{W}_{rij} \mathcal{W}_{rkl} \mathbf{E}'_{jl} \tag{9}$$

where $\mathbf{E}' = \mathbf{E}^\top \mathbf{E} \in \mathbb{R}^{d \times d}$. The complexity of computing $Z$ depends on the order of tensor contractions in the einsum operation showed in Eq. (9). By optimising the order of tensor contractions (e.g., by using software libraries like opt_einsum), we retrieve that computing $Z$ requires time $\mathcal{O}(|\mathcal{E}| \cdot d^2 + |\mathcal{R}| \cdot d^3)$ and additional space $\mathcal{O}(|\mathcal{R}| \cdot d^2)$. Notice that the time complexity here is slightly lower than the theoretical upper bound given in Thm. 1, which would be $\mathcal{O}(|\mathcal{E}| \cdot d^2 + |\mathcal{R}| \cdot d^4)$.

**TUCKER².** Lastly, we present the derivation of the partition function for TUCKER$^2$. The score function $\phi_{\mathrm{TUCKER}^2}$ encoded by TUCKER$^2$ can be written as

$$\phi_{\mathrm{TUCKER}^2}(s, r, o) = \left( \mathcal{T} \times_1 \mathbf{e}_s \times_2 \mathbf{w}_r \times_3 \mathbf{e}_o \right)^2$$
$$= \sum_{\substack{(i,j,k) \in \\ [d_e] \times [d_r] \times [d_e]}} \sum_{\substack{(l,m,n) \in \\ [d_e] \times [d_r] \times [d_e]}} \tau_{ijk} \tau_{lmn} e_{si} e_{sl} w_{rj} w_{rm} e_{ok} e_{on}$$

where $\mathbf{e}_s, \mathbf{e}_o \in \mathbb{R}^{d_e}$ are rows of the matrix $\mathbf{E} \in \mathbb{R}^{|\mathcal{E}| \times d_e}$, $\mathbf{w}_r$ is a row of the matrix $\mathbf{W} \in \mathbb{R}^{|\mathcal{R}| \times d_r}$, and $\mathcal{T} \in \mathbb{R}^{d_e \times d_r \times d_e}$ denotes the core tensor. The partition function of $\phi_{\mathrm{TUCKER}^2}$ can be written as

$$Z = \sum_{s \in \mathcal{E}} \sum_{r \in \mathcal{R}} \sum_{o \in \mathcal{E}} \phi_{\mathrm{TUCKER}^2}(s, r, o)$$
$$= \sum_{\substack{(i,j,k) \in \\ [d_e] \times [d_r] \times [d_e]}} \sum_{\substack{(l,m,n) \in \\ [d_e] \times [d_r] \times [d_e]}} \tau_{ijk} \tau_{lmn} \left( \sum_{s \in \mathcal{E}} e_{si} e_{sl} \right) \left( \sum_{r \in \mathcal{R}} w_{rj} w_{rm} \right) \left( \sum_{o \in \mathcal{E}} e_{ok} e_{on} \right)$$
$$= \mathcal{T}_{ijk} \mathcal{T}_{lmn} \mathbf{E}'_{il} \mathbf{W}'_{jm} \mathbf{E}'_{kn} \tag{10}$$

where $\mathbf{E}' = \mathbf{E}^\top \mathbf{E} \in \mathbb{R}^{d_e \times d_e}$ and $\mathbf{W}' = \mathbf{W}^\top \mathbf{W} \in \mathbb{R}^{d_r \times d_r}$. Similarly to RESCAL$^2$, by optimising the order of tensor contractions in the einsum operation showed in Eq. (10), we retrieve that computing $Z$ requires time $\mathcal{O}(|\mathcal{E}| \cdot d_e^2 + |\mathcal{R}| \cdot d_r^2 + d_e^2 \cdot d_r)$ and additional space $\mathcal{O}(d_e^2 \cdot d_r)$. Similarly to RESCAL$^2$, the time complexity is lower than the theoretical upper bound given in Thm. 1, which would be $\mathcal{O}(|\mathcal{E}| \cdot d_e^2 + |\mathcal{R}| \cdot d_r^2 + d_e^4 \cdot d_r^2)$.

### C.4.2 Complexity of Computing the PLL Objective

In this section, we show that GeKCs enable to better scale the computation of the PLL objective (Eq. (1)) with respect to energy-based KGE models (see Section 2). We present this concept for CP and GeKCs derived from it (Section 4), as for the other score functions it is similar.

**Complexity of the PLL objective on CP.** Let $\phi_{\mathrm{CP}}(s,r,o) = \langle \mathbf{e}_s, \mathbf{w}_r, \mathbf{e}_o \rangle = \sum_{i=1}^d e_{si}w_{ri}e_{oi}$ be the score function of CP [40], where $\mathbf{e}_s, \mathbf{e}_o \in \mathbb{R}^d$ (resp. $\mathbf{w}_r \in \mathbb{R}^d$) are rows of the matrices $\mathbf{U}, \mathbf{V} \in \mathbb{R}^{|\mathcal{E}|\times d}$ (resp. $\mathbf{W} \in \mathbb{R}^{|\mathcal{R}|\times d}$), which associate to each entity (resp. predicate) a vector. Given a training triple $(s,r,o)$, the computation of the term $\log p(o \mid s, r) = \phi(s,r,o) - \log \sum_{o' \in \mathcal{E}} \exp \phi(s,r,o')$ requires evaluating $\phi_{\mathrm{CP}}(s,r,o')$ for all objects $o' \in \mathcal{E}$. In order to fully exploit GPU parallelism [36], this is usually done with the matrix-vector multiplication $\mathbf{V}(\mathbf{e}_s \odot \mathbf{w}_r) \in \mathbb{R}^{|\mathcal{E}|}$, where $\odot$ denotes the Hadamard product [40, 12]. Therefore, computing $\log p(o \mid s, r)$ for each triple $(s,r,o)$ in a mini-batch $B \subset \mathcal{E} \times \mathcal{R} \times \mathcal{E}$ such that $|\mathcal{E}| \gg |\mathcal{B}|$ requires time $\mathcal{O}(|\mathcal{E}| \cdot |B| \cdot d)$ and space $\mathcal{O}(|\mathcal{E}| \cdot |B|)$. For the other terms of the PLL objective (i.e., $\log p(s \mid r, o)$ and $\log p(r \mid s, o)$) the derivation is similar. Moreover, for real-world large KGs it is reasonable to assume that $|\mathcal{E}| \gg |\mathcal{R}|$ and therefore the cost of computing $\log p(r \mid s, o)$ is negligible.

**Complexity of the PLL objective on GeKCs.** GeKCs obtained from CP either by non-negative restriction (Section 4.1) or by squaring (Section 4.2) encode $\phi_{\mathsf{pc}}(s,r,o) \propto p(s,r,o)$ for any input triple. As such, the component $\log p(o \mid s, r)$ of the PLL objective can be written as

$$\log p(o \mid s, r) = \log \phi_{\mathsf{pc}}(s,r,o) - \log \sum_{o' \in \mathcal{E}} \phi_{\mathsf{pc}}(s,r,o'). \tag{11}$$

The absence of the exponential function in the summed terms in Eq. (11) allows us to push the outer summation inside the circuit computing $\phi_{\mathsf{pc}}(s,r,o)$, and to sum over the input units relative to objects. For instance, for CP⁺ we can write

$$\sum_{o' \in \mathcal{E}} \phi_{\mathrm{CP}^+}(s,r,o') = \sum_{o' \in \mathcal{E}} \sum_{i=1}^d e_{si}w_{ri}e_{oi} = \sum_{i=1}^d e_{si}w_{ri}\left(\sum_{o \in \mathcal{E}} e_{o'i}\right) = (\mathbf{e}_s \odot \mathbf{w}_r)^\top (\mathbf{V}^\top \mathbf{1}_\mathcal{E})$$

where $\mathbf{e}_s, \mathbf{w}_r, \mathbf{e}_o \in \mathbb{R}_+^d$, $\mathbf{V} \in \mathbb{R}_+^{|\mathcal{E}|\times d}$ denotes the matrix whose rows are object embeddings, and $\mathbf{1}_\mathcal{E} = [1 \ldots 1]^{|\mathcal{E}|}$ is a vector of ones. Note that $\mathbf{V}^\top \mathbf{1}_\mathcal{E} \in \mathbb{R}_+^d$ does not depend on the input triple. Therefore, given a mini-batch of triples $B$, computing $\log p(o \mid s, r)$ requires time $\mathcal{O}((|\mathcal{E}| + |B|) \cdot d)$ and space $\mathcal{O}(|B| \cdot d)$, which is much lower than the complexity on CP showed above, and we can still leverage GPU parallelism. For CP², the complexity is similar to the derivation of the partition function complexity showed in Appendix C.4.1. That is, for CP² we can write

$$\sum_{o' \in \mathcal{E}} \phi_{\mathrm{CP}^2}(s,r,o') = \sum_{o' \in \mathcal{E}} \left(\sum_{i=1}^d e_{si}w_{ri}e_{oi}\right)^2 = \sum_{i=1}^d \sum_{j=1}^d e_{si}e_{sj}w_{ri}w_{rj}\left(\sum_{o' \in \mathcal{E}} e_{o'i}e_{o'j}\right)$$
$$= (\mathbf{e}_s \odot \mathbf{w}_r)^\top (\mathbf{V}^\top \mathbf{V})(\mathbf{e}_s \odot \mathbf{w}_r)$$

where $\mathbf{e}_s, \mathbf{w}_r, \mathbf{e}_o \in \mathbb{R}^d$, $\mathbf{V} \in \mathbb{R}^{|\mathcal{E}|\times d}$. Note that the matrix $\mathbf{V}^\top \mathbf{V} \in \mathbb{R}^{d\times d}$ does not depend on the input triple. Therefore, given a mini-batch of triples $B$, computing $\log p(o \mid s, r)$ requires time $\mathcal{O}((|\mathcal{E}| + |B|) \cdot d^2)$ and space $\mathcal{O}(|B| \cdot d)$. While the time complexity is quadratic in the embedding size $d$, it is still much lower than the time complexity on CP. A similar discussion can also be carried out for the other KGE models and the corresponding GeKCs, which retrieves the complexities showed in Table C.1.

### C.4.3 Training Speed-up Benchmark Details

In this section we report the details about the training benchmark on COMPLEX, COMPLEX⁺ and COMPLEX², whose results are showed in Fig. 3. We measure time and peak GPU memory usage required for computing the PLL objective (Eq. (1)) *and* to do an optimisation step for a single batch on ogbl-wikikg2 [32], a large knowledge graph with millions of entities (see Table F.1). We fix the embedding size to $d = 100$ for the three models. For the benchmark with increasing batch size, we keep all the entities and increase the batch size from 100 to 5000. For the benchmark with increasing number of entities, we keep the batch size fixed to 500 (the maximum allowed for COMPLEX by our GPUs) and progressively increase the number of entities, from about $3 \cdot 10^5$ to $2.5 \cdot 10^6$. We report the average time over 25 independent runs on a single Nvidia RTX A6000 with 48 GiB of memory.

# D  Distribution of Scores

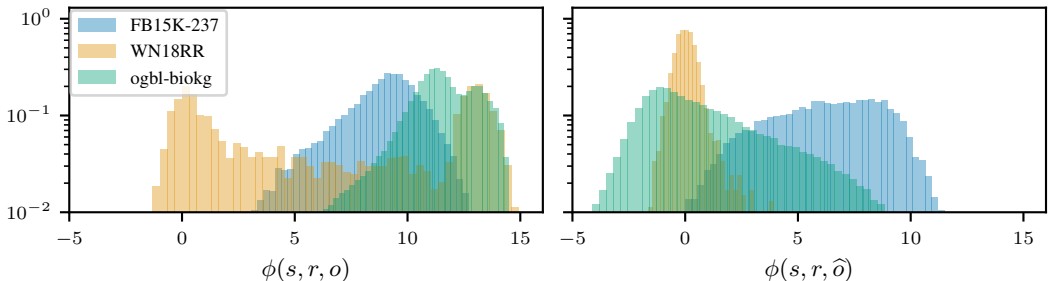

Figure D.1: **Scores are mostly non-negative.** Histograms of the scores assigned by COMPLEX to existing validation triples (left) and their perturbation (right) on three data sets. The vast majority of triple scores are non-negative, suggesting that squaring them has minimal effect on the rankings.

In Fig. D.1 we show the histograms of the scores assigned to the validation triples and their perturbations of three data sets (see Appendix F.1). Following Socher et al. [59], we generate triple perturbations that are challenging for link prediction. That is, for each validation triple $(s, r, o)$, the corresponding perturbation $(s, r, \widehat{o})$ is obtained by replacing the object with a random entity that has appeared at least once as an object in a training triple with predicate $r$. The bottom line is that scores are mostly non-negative, and hence can be used as a heuristic to effectively initialise GeKCs or quickly distil them (e.g., on FB15K-237), as we further discuss in Appendix F.5.1.

# E  Reconciling Knowledge Graph Embeddings Interpretations

**Triples as boolean variables.**  KGE models such as CP, RESCAL, TUCKER and COMPLEX have been historically introduced as factorizations of a tensor-representation of a KG, which we discuss next. In fact, a KG $\mathcal{G}$ can be represented as a 3-way binary tensor $\mathbf{Y} \in \{0, 1\}^{|\mathcal{E}| \times |\mathcal{R}| \times |\mathcal{E}|}$ in which every entry $Y_{sro}$ is 1 if $(s, r, o) \in \mathcal{G}$ and 0 otherwise [48]. Under this light, a KGE model like RESCAL factorises every slice $\mathbf{Y}_r \in \{0, 1\}^{|\mathcal{E}| \times |\mathcal{E}|}$, corresponding to the predicate $r$ as $\mathbf{E}\mathbf{W}_r\mathbf{E}^\top$ where $\mathbf{E}$ is an $|\mathcal{E}| \times d$ matrix comprising the entity embeddings and $\mathbf{W}_r$ is an $d \times d$ matrix containing the embeddings for the predicate $r$. To deal with uncertainty and incomplete KGs, $Y_{sro}$ can be interpreted as a Bernoulli random variable. As such, its distribution becomes $p(Y_{sro} = 1 \mid s, r, o)$ which is usually modelled as $\sigma(\phi(s, r, o))$, where $\sigma$ denotes the logistic function and $\phi$ is the score function of a KGE model. Note that this distribution of triple introduces $|\mathcal{E} \times \mathcal{R} \times \mathcal{E}|$ random variables, one for each possible triple.

**KGE models as estimators of a distribution over KGs.**  At the same time, the interpretation of triples as boolean variables *induces a distribution over possible KGs*, $q(\mathcal{G})$, which is the distribution over all possible binary tensors $p(\mathbf{Y})$. The probability of a KG $\mathcal{G}$ can therefore be computed as the product of the likelihoods of all variables $Y_{sro}$, i.e., $q(\mathcal{G}) = \prod_{(s,r,o) \in \mathcal{G}} p(Y_{sro} = 1 \mid s, r, o) \cdot \prod_{(s,r,o) \notin \mathcal{G}} p(Y_{sro} = 0 \mid s, r, o)$. Note that (re-)normalising this distribution is intractable in general, as it would require summing over all possible $2^{|\mathcal{E} \times \mathcal{R} \times \mathcal{E}|}$ binary tensors. This is why historically KGE models have been interpreted as energy-based models, by directly optimising for $\phi(s, r, o)$, interpreted as the negated energy associated to every triple, and not $p(Y_{sro} = 1 \mid s, r, o)$ (see Section 2). This has been done via negative sampling or other contrastive learning objectives [6, 7]. We point out that this very same interpretation can be found in the literature of probabilistic logic programming [25], probabilistic databases (PDBs) [14] and statistical relational learning (see Section 6) where the distribution over possible "worlds" is over sets of boolean assignments to ground atoms or facts, or tuples in a PDB, each interpreted as Bernoulli random variables.

**Estimating a distribution over triples.**  In this work, instead, we interpret existing KGE models and our GeKCs as models that encode a possibly unnormalised probability distribution over three random variables, $S, R, O$, which induces a distribution over triples that is tractable to renormalise.[7]

---

[7]The polynomial cost of renormalising an energy-based KGE is unfortunately infeasible for real-world KGs, see Section 2.

To reconcile these two perspectives, we interpret the probability of a triple $p(s, r, o)$ to be proportional to the probability of all KGs $\mathcal{G}$ where $(s, r, o)$ holds, i.e., those $\mathcal{G}$ such that $(s, r, o) \in \mathcal{G}$. Intuitively, a triple will be more probable to exist if it does exist in highly probable KGs. More formally, given $q$ a probability distribution over KGs, we define $p$ as an unnormalised probability distribution over triples, i.e., $\mu(s, r, o) \propto p(s, r, o)$, where

$$\mu(s, r, o) = \sum_{\substack{\mathcal{G} \in \mathcal{H} \\ (s,r,o) \in \mathcal{G}}} q(\mathcal{G}) = \sum_{\mathcal{G} \in \mathcal{H}} q(\mathcal{G}) \cdot \mathbb{1}\{(s, r, o) \in \mathcal{G}\} = \mathbb{E}_{\mathcal{G} \sim q}[\mathbb{1}\{(s, r, o) \in \mathcal{G}\}] \quad (12)$$

and $\mathcal{H} = 2^{\mathcal{E} \times \mathcal{R} \times \mathcal{E}}$ denotes the set of all possible KGs. Computing the expectation in Eq. (12) exactly is equivalent to solving a *weighted model counting* (WMC) problem [10], where we sum the probabilities of all possible KGs containing $(s, r, o)$. Alternatively, it is equivalent to computing the probability of the simplest possible query in a PDB (i.e., asking for a single tuple), where each stored tuple is interpreted as an independent Bernoulli random variable $Y_{sro}$. Therefore, we have that $\mu(s, r, o)$ is simply the likelihood that $Y_{sro}$ is true, i.e., $p(Y_{sro} = 1 \mid s, r, o)$. Furthermore, the normalisation constant of $\mu$ (Eq. (12)) can be written as

$$Z = \sum_{\substack{(s,r,o) \in \\ \mathcal{E} \times \mathcal{R} \times \mathcal{E}}} \mu(s, r, o) = \sum_{\mathcal{G} \in \mathcal{H}} q(\mathcal{G}) \cdot \sum_{\substack{(s,r,o) \in \\ \mathcal{E} \times \mathcal{R} \times \mathcal{E}}} \mathbb{1}\{(s, r, o) \in \mathcal{G}\} = \sum_{\mathcal{G} \in \mathcal{H}} q(\mathcal{G}) \cdot |\mathcal{G}| = \mathbb{E}_{\mathcal{G} \sim q}[|\mathcal{G}|]$$

which is the expected size of a KG according to the probability distribution $q$. Written in this way, however, computing $Z$ through $q(\mathcal{G})$ is intractable. For this reason, we directly encode $\mu$ with GeKCs and compute $Z$ by summing over all triples, and therefore without modelling $q(\mathcal{G})$.

**Further interpretations in related works.** Under the interpretation of a KG as a PDB, Friedman and Van den Broeck [26] further decompose the likelihood that $Y_{sro}$ is true as

$$p(Y_{sro} = 1 \mid s, r, o) = p(E_s = 1 \mid s) \cdot p(T_r = 1 \mid r) \cdot p(E_o = 1 \mid o)$$

where $E_s, E_o, T_r$ are new Bernoulli variables that are assumed to be conditionally independent given the parameters of the PDB. That is, instead of introducing one random variable per triple, they introduce one random variable per entity and predicate. In this framework, they reinterpret the score function of DISTMULT, a simplified variant of CP, as an implicit circuit that models an unnormalized distribution over the collection of variables $\mathbf{Z} = \{E_u\}_{u \in \mathcal{E}} \cup \{T_r\}_{r \in \mathcal{R}}$, trained by negative sampling. This decomposition permits to compute the probability of any database query efficiently, which otherwise is known to be either a PTIME or a #P-hard problem, depending on the query type [15]. If we were to interpret our distribution $\mu(S = s, R = r, O = o)$ as the unnormalized marginal distribution $p(E_s = 1, T_r = 1, E_o = 1) = \sum_{\mathbf{z}'} p(E_s = 1, T_r = 1, E_o = 1, \mathbf{Z}' = \mathbf{z}')$, where $\mathbf{Z}' = \mathbf{Z} \setminus \{E_s, T_r, E_o\}$, we could equivalently compute any probabilistic query efficiently. Note that under this interpretation, training our GeKCs by MLE over $S, R, O$ would be equivalent to maximise a composite marginal log-likelihood [70] over $\mathbf{Z}$.

## F    Empirical Evaluation

### F.1    Datasets Statistics

Table F.1 shows statistics of commonly-used datasets to benchmark KGE models for link prediction. We employ standard benchmark datasets [62, 21, 32] whose number of entities (resp. predicates) ranges from $\approx$14k to $\approx$2.5M (resp. from 11 to $\approx$500).

Table F.1: **Dataset statistics.** Statistics of multi-relational knowledge graphs: number of entities ($|\mathcal{E}|$), number of predicates ($|\mathcal{R}|$), number of training/validation/test triples.

| Dataset | | $|\mathcal{E}|$ | $|\mathcal{R}|$ | # Train | # Valid | # Test |
|---|---|---|---|---|---|---|
| FB15k-237 | [62] | 14,541 | 237 | 272,115 | 17,535 | 20,466 |
| WN18RR | [21] | 40,943 | 11 | 86,835 | 3,034 | 3,134 |
| ogbl-biokg | [32] | 93,773 | 51 | $4{,}763 \cdot 10^3$ | $163 \cdot 10^3$ | $163 \cdot 10^3$ |
| ogbl-wikikg2 | [32] | $2.5 \cdot 10^6$ | 535 | $16{,}109 \cdot 10^3$ | $429 \cdot 10^3$ | $598 \cdot 10^3$ |

## F.2 Metrics

**Mean reciprocal rank and hits at** $k$**.** Given a test triple $(s, r, o)$, we rank the possible object $o'$ (resp. subject $s'$) completions to link prediction queries $(s, r, ?)$ (resp. $(?, r, o)$) based on their scores in descending order. The position of the test triple $(s, r, o)$ in the ranking of object (resp. subject) completed queries $(s, r, o')$ (resp. $(s', r, o)$) is then used to compute the *mean reciprocal rank* (MRR)

$$\text{MRR} = \frac{1}{2|\mathcal{G}_{\text{test}}|} \sum_{(s,r,o)\in\mathcal{G}_{\text{test}}} \left( \frac{1}{\text{rank}(o \mid s, r)} + \frac{1}{\text{rank}(s \mid r, o)} \right)$$

where $\mathcal{G}_{\text{test}}$ denotes the set of test triples, and $\text{rank}(o \mid s, r), \text{rank}(s \mid r, o)$ denote respectively the positions of the true completion $(s, r, o)$ in the rankings of object and subject completed queries. The fraction of *hits at* $k$ (Hits@$k$) for $k > 0$ is computed as

$$\text{Hits@}k = \frac{1}{2|\mathcal{G}_{\text{test}}|} \sum_{(s,r,o)\in\mathcal{G}_{\text{test}}} \left( \mathbb{1}\{\text{rank}(o \mid s, r) \le k\} + \mathbb{1}\{\text{rank}(s \mid r, o) \le k\} \right).$$

Consistently with existing works on link prediction [56, 12], the MRRs and Hits@$k$ metrics are computed under the *filtered* setting, i.e., we rank true completed triples against potential ones that do not appear in the union of training, validation and test splits.

**Semantic consistency score.** Let $K$ be a logical constraint encoding some background knowledge over variables $S$, $R$ and $O$. Given a test triple $(s, r, o)$, we first rank the possible completions to link prediction queries in the same way as for computing the MRR. Then, the *semantic consistency score* (Sem@$k$) [33] for some integer $k > 0$ is computed as

$$\text{Sem@}k = \frac{1}{2k|\mathcal{G}_{\text{test}}|} \sum_{(s,r,o)\in\mathcal{G}_{\text{test}}} \left( \sum_{o'\in\mathcal{A}_O^k(s,r,o)} \mathbb{1}\{(s, r, o') \models K\} + \sum_{s'\in\mathcal{A}_S^k(s,r,o)} \mathbb{1}\{(s', r, o) \models K\} \right)$$

where $\mathcal{G}_{\text{test}}$ denotes the set of test triples, $\mathcal{A}_O^k(s, r, o)$ (resp. $\mathcal{A}_S^k(s, r, o)$) denotes the list of the top-$k$ candidate object (resp. subject) completions to the link prediction query $(s, r, ?)$ (resp. $(?, r, o)$), and $(s, r, o) \models K$ if and only if $(s, r, o)$ satisfies $K$.

## F.3 Empirical KTD Score

Let $\mathcal{F} = \{x_i\}_{i=1}^m$, $\mathcal{G} = \{y_j\}_{j=1}^n$ two sets of triples that are drawn i.i.d. from two distributions $\mathbb{P}, \mathbb{Q}$ over triples. We compute the empirical KTD score with an *unbiased estimator* [28] $\text{KTD}_u(\mathcal{F}, \mathcal{G})$ as

$$\frac{1}{m(m-1)} \sum_{i \neq j}^m k(\psi(x_i), \psi(x_j)) + \frac{1}{n(n-1)} \sum_{i \neq j}^n k(\psi(y_i), \psi(y_j)) - \frac{2}{mn} \sum_{i=1}^m \sum_{j=1}^n k(\psi(x_i), \psi(y_j)).$$

For each data set, we compute the empirical KTD score between $n = 25,000$ triples sampled from GeKCs and $m$ test triples. In case of $m > n$, we sample $n$ triples randomly, uniformly and without replacement from the set of test triples. The time complexity of computing the KTD score is $\mathcal{O}(nmh)$, where $h$ denotes the size of triple latent representations ($h = 4000$ in our case, see Section 7.3). For efficiency reasons, we therefore follow Binkowski et al. [4] and randomly extract two batches of 1000 triples each from both the generated and the test triples sets and compute the empirical KTD score on them. We repeat this process 100 times and report the average and standard deviation in Table F.6.

## F.4 Experimental Setting

**Hyperparameters.** All models are trained by gradient descent with either the PLL or the MLE objective (Eqs. (1) and (2)). We set the weights $\omega_s, \omega_r, \omega_o$ of the PLL objective all to one, as to retrieve a classical pseudo-log-likelihood [70]. Note that Chen et al. [12] set $\omega_s, \omega_o$ to one and treat $\omega_r$ as an additional hyperparameter instead that is opportunely tuned. The models are trained until the MRR computed on the validation set does not improve after three consecutive epochs. We fix the embedding size $d = 1000$ for both CP and COMPLEX and use Adam [38] as optimiser with $10^{-3}$ as learning rate. An exception is made for GeKCs obtained via non-negative restriction (Section 4.1), for which a learning rate of $10^{-2}$ is needed, as we observed very slow convergence rates. We search for the batch size in $\{5 \cdot 10^2, 10^3, 2 \cdot 10^3, 5 \cdot 10^3\}$ based on the validation MRR. Finally, we perform 5 repetitions with different seeds and report the average MRR and two standard deviations in Table F.2.

**Parameters initialisation.** Following [40, 12], the parameters of CP and COMPLEX are initialised by sampling from a normal distribution $\mathcal{N}(0, \sigma^2)$ with $\sigma = 10^{-3}$. Since the embedding values in $\text{CP}^+$ and $\text{COMPLEX}^+$ can be interpreted as parameters of categorical distributions over entities and predicates (see Appendix C.1), we initialise them by sampling from a Dirichlet distribution with concentration factors set to $10^3$. To allow unconstrained optimisation for $\text{CP}^+$ and $\text{COMPLEX}^+$, we represent the embedding values by their logarithm and perform computations directly in log-space, i.e., summations and log-sum-exp operations instead of multiplications and summations, respectively. Moreover, the parameters that ensure the non-negativity of $\text{COMPLEX}^+$ (see Appendix C.2) are initialised by sampling from a normal distribution $\mathcal{N}(0, \sigma^2)$ with $\sigma = 10^{-2}$. We initialise the parameters of $\text{CP}^2$ and $\text{COMPLEX}^2$ such that the logarithm of the scores are approximately normally distributed and centred in zero during the initial optimisation steps, since this applies for the scores given by CP and COMPLEX. Such initialisation therefore permits a fairer comparison. To do so, we initialise the embedding values by sampling from a log-normal distribution $\mathcal{LN}(\mu, \sigma^2)$, where $\mu = -\log(d)/3 - \sigma^2/2$ for CP and $\mu = -\log(2d)/3 - \sigma^2/2$ for COMPLEX, both with $\sigma = 10^{-3}$. The mentioned values for $\mu$ can be derived via Fenton-Wilkinson approximation [24]. Even though the parameters of GeKCs obtained via non-monotonic squaring are initialised to be non-negative, they are free of becoming negative during training (as we also confirm in practice).

**Hardware.** Experiments on the smaller knowledge graphs FB15K-237 and WN18RR were run on a single Nvidia GTX 1060 with 6 GiB of memory, while those on the larger ogbl-biokg and ogbl-wikikg2 were run on a single Nvidia RTX A6000 with 48 GiB of memory.

### F.5 Additional Experimental Results

#### F.5.1 Link Prediction Results

In this section, we present the additional results regarding the link prediction experiments showed in Section 7.1 and analyse different metrics and learning settings.

**Statistical tests and hits at $k$.** Table F.2 shows the best test average MRRs (see Appendix F.2) with two standard deviation and average training time across 5 independent runs with different seeds. We highlight the best results in bold according to a one-sided Mann–Whitney U test with a confidence level of 99%. The showed results in terms of MRRs are also confirmed in Table F.3, which shows the best average Hits@$k$ (see Appendix F.2) with $k \in \{1, 3, 10\}$.

**Average log-likelihood.** For the best GeKCs for link prediction showed in Table F.2, we report the average log-likelihood of test triples and two standard deviations (across 5 independent runs) in Table F.4. We again highlight the best results in bold, according to a one-sided Mann–Whitney U test.

**Quickly distilling parameters.** As discussed in Section 4.2, since learned KGE models mostly assign non-negative scores to triples (see Appendix D) we can initialise the parameters of GeKCs obtained by squaring with the parameters of already-learned KGE models, without losing much in terms of link prediction performances. Here, we test this hypothesis and fine-tune GeKCs initialised in this way by using either the PLL or MLE objectives (Eqs. (1) and (2)). To do so, we first collect the parameters of the best CP and COMPLEX found for link prediction (see Section 7.1). Then, we initialise GeKCs derived by squaring with these parameters and fine-tune them until the MRR computed on validation triples does not improve after three consecutive epochs. We employ Adam [38] as optimiser, and we search for the batch size in $\{5 \cdot 10^2, 10^3, 2 \cdot 10^3, 5 \cdot 10^3\}$ and learning rate in $\{10^{-3}, 10^{-4}\}$, as fine-tuning may require a lower learning rate than the one used in previous experiments (see Appendix F.4). Table F.5 shows the MRRs achieved by CP, COMPLEX and the corresponding GeKCs obtained via squaring that are initialised by distilling the parameters from the already-trained CP and COMPLEX. On FB15K-237 and WN18RR, distilling parameters induces a substantial improvement in terms of MRR with respect to $\text{CP}^2$ and $\text{COMPLEX}^2$ whose parameters have been initialised randomly (see Appendix F.4). Furthermore, for $\text{COMPLEX}^2$ and on WN18RR and ogbl-biokg we achieved similar MRRs with respect to COMPLEX *without* the need of fine-tuning.

Table F.2: **GeKCs are competitive with their energy-based counterparts.** Best test MRRs (and two standard deviations) of CP, COMPLEX and GeKCs trained with the PLL and MLE objectives (Eqs. (1) and (2)). In parentheses we show the average training time (in minutes).

| Model | FB15k-237 | | WN18RR | | ogbl-biokg | |
|---|---|---|---|---|---|---|
| | PLL | MLE | PLL | MLE | PLL | MLE |
| CP | 0.310 ±0.001 (8) | — | **0.105 ±0.007** (11) | — | 0.831 ±0.001 (136) | — |
| CP$^+$ | 0.237 ±0.003 (1) | 0.230 ±0.003 (1) | 0.027 ±0.002 (1) | 0.026 ±0.001 (1) | 0.496 ±0.013 (172) | 0.501 ±0.010 (142) |
| CP$^2$ | **0.315 ±0.003** (8) | 0.282 ±0.004 (7) | **0.104 ±0.001** (23) | 0.091 ±0.004 (23) | **0.848 ±0.001** (66) | 0.829 ±0.001 (61) |
| COMPLEX | **0.342 ±0.005** (36) | — | **0.471 ±0.002** (16) | — | 0.829 ±0.001 (180) | — |
| COMPLEX$^+$ | 0.214 ±0.003 (10) | 0.205 ±0.006 (5) | 0.030 ±0.001 (6) | 0.029 ±0.001 (3) | 0.503 ±0.014 (245) | 0.516 ±0.009 (212) |
| COMPLEX$^2$ | 0.334 ±0.001 (10) | 0.300 ±0.003 (16) | 0.420 ±0.011 (37) | 0.391 ±0.004 (19) | **0.858 ±0.001** (71) | 0.840 ±0.001 (59) |

Table F.3: **Hits@$k$ results.** Average test Hits@$k$ for $k \in \{1, 3, 10\}$ of CP, COMPLEX and CP$^2$ and COMPLEX$^2$ trained with the PLL or MLE objectives.

| | FB15k-237 | | | | | | WN18RR | | | | | | ogbl-biokg | | | | | |
|---|---|---|---|---|---|---|---|---|---|---|---|---|---|---|---|---|---|---|
| | PLL | | | MLE | | | PLL | | | MLE | | | PLL | | | MLE | | |
| Model $k =$ | 1 | 3 | 10 | 1 | 3 | 10 | 1 | 3 | 10 | 1 | 3 | 10 | 1 | 3 | 10 | 1 | 3 | 10 |
| CP | 22.4 | 34.1 | 48.2 | — | — | — | 7.5 | 12.1 | 16.8 | — | — | — | 76.4 | 88.1 | 95.0 | — | — | — |
| CP$^+$ | 17.0 | 25.8 | 36.7 | 16.7 | 24.9 | 35.4 | 1.7 | 2.7 | 4.5 | 1.6 | 2.5 | 4.4 | 38.0 | 54.4 | 73.4 | 38.4 | 55.0 | 74.4 |
| CP$^2$ | 23.1 | 34.8 | 48.2 | 20.5 | 30.8 | 43.5 | 6.7 | 12.1 | 17.6 | 5.9 | 10.7 | 15.3 | 78.6 | 89.5 | 95.7 | 76.1 | 88.1 | 95.0 |
| COMPLEX | 25.2 | 37.5 | 52.5 | — | — | — | 43.3 | 48.6 | 54.6 | — | — | — | 76.1 | 87.9 | 95.0 | — | — | — |
| COMPLEX$^+$ | 15.7 | 23.1 | 31.7 | 15.0 | 22.1 | 30.4 | 1.5 | 2.7 | 4.5 | 1.6 | 2.5 | 4.4 | 38.8 | 55.1 | 74.1 | 40.0 | 56.7 | 75.9 |
| COMPLEX$^2$ | 24.5 | 36.9 | 51.1 | 21.6 | 33.0 | 46.7 | 36.0 | 45.6 | 52.4 | 34.5 | 42.3 | 46.9 | 80.0 | 90.1 | 95.8 | 77.5 | 88.8 | 95.4 |

Table F.4: **Better distribution estimation with GeKCs obtained via squaring.** Average log-likelihood of test triples achieved by baselines and GeKCs trained with the PLL or MLE objectives.

| Model | FB15k-237 | | WN18RR | | ogbl-biokg | |
|---|---|---|---|---|---|---|
| Uniform | -24.638 | | -23.638 | | -26.829 | |
| NNMFAug | -19.270 | | -22.938 | | -17.562 | |
| | PLL | MLE | PLL | MLE | PLL | MLE |
| CP$^+$ | -16.773 ±0.040 | -16.592 ±0.059 | **-21.987 ±0.006** | -22.103 ±0.010 | -17.900 ±0.048 | -17.416 ±0.049 |
| CP$^2$ | -17.105 ±0.031 | **-15.982 ±0.028** | -24.911 ±0.241 | -26.352 ±0.077 | -17.231 ±0.059 | **-16.533 ±0.013** |
| COMPLEX$^+$ | -17.507 ±0.035 | -17.592 ±0.039 | -21.233 ±0.058 | -21.432 ±0.008 | -18.716 ±0.088 | -17.749 ±0.019 |
| COMPLEX$^2$ | -17.100 ±0.026 | **-15.744 ±0.041** | **-19.522 ±0.530** | **-19.739 ±0.214** | -17.340 ±0.022 | **-16.518 ±0.003** |

Table F.5: **Distilling parameters can improve performances.** Test MRRs achieved by CP, COMPLEX and GeKCs obtained by squaring (Section 4.2). For CP$^2$ and COMPLEX$^2$ we report the best MRRs achieved by distilling the parameters from the already-learned CP and COMPLEX (denoted with $\star$), and with † we denote those results for which further fine-tuning with the PLL or MLE objectives did not bring better results. We underline results for which distilling parameters increased the MRR.

| Model | FB15k-237 | | WN18RR | | ogbl-biokg | |
|---|---|---|---|---|---|---|
| | PLL | MLE | PLL | MLE | PLL | MLE |
| CP | 0.311 | — | 0.108 | — | 0.831 | — |
| COMPLEX | 0.344 | — | 0.470 | — | 0.829 | — |
| CP$^2$ | 0.317 | 0.285 | 0.103 | 0.089 | 0.849 | 0.830 |
| CP$^2$ $\star$ | 0.327 | 0.315 | 0.102 | 0.115 | 0.851 | 0.828 † |
| COMPLEX$^2$ | 0.333 | 0.301 | 0.416 | 0.390 | 0.859 | 0.839 |
| COMPLEX$^2$ $\star$ | 0.342 | 0.340 | 0.462 † | 0.463 | 0.859 | 0.828 † |

### F.5.2 Quality of Sampled Triples Results

In this section, we provide additional results regarding the evaluation of the quality of triples sampled by GeKCs (see Section 7.3). For these experiments, we search for the same hyperparameters as for the experiments on link prediction (see Appendix F.4), and train GeKCs until the average log-likelihood computed on validation triples does not improve after three consecutive epochs.

Table F.6 shows the mean empirical KTD score and one standard deviation (see Appendix F.3). In addition, we visualise triple embeddings of sampled and test triples in Fig. F.1 by leveraging t-SNE [69] as a method for visualising high-dimensional data. In particular, we apply the t-SNE method implemented in scikit-learn with perplexity 50 and number of iterations $5 \cdot 10^3$, while other parameters are fixed to their default value. As showed in Fig. F.1c, an empirical KTD score close to zero translates to an high clusters similarity between embeddings of sampled and test triples.

Table F.6: **GeKCs trained by MLE generate new likely triples.** Empirical KTD scores between test triples and triples generated by baselines and GeKCs trained with the PLL objective or by MLE (Eqs. (1) and (2)). Lower is better.

| Model | FB15k-237 | | WN18RR | | ogbl-biokg | |
|---|---|---|---|---|---|---|
| Training set | 0.055 ±0.007 | | 0.260 ±0.013 | | 0.029 ±0.010 | |
| Uniform | 0.589 ±0.012 | | 0.766 ±0.036 | | 1.822 ±0.044 | |
| NNMFAug | 0.414 ±0.014 | | 0.607 ±0.028 | | 0.518 ±0.035 | |
| | PLL | MLE | PLL | MLE | PLL | MLE |
| CP$^+$ | 0.404 ±0.016 | 0.433 ±0.015 | 0.633 ±0.033 | **0.578 ±0.029** | 0.966 ±0.040 | 0.738 ±0.030 |
| CP$^2$ | 0.253 ±0.014 | **0.070 ±0.007** | 0.768 ±0.036 | 0.768 ±0.036 | 0.039 ±0.009 | **0.017 ±0.013** |
| COMPLEX$^+$ | 0.336 ±0.016 | 0.323 ±0.015 | 0.456 ±0.018 | 0.478 ±0.019 | 0.175 ±0.019 | 0.097 ±0.013 |
| COMPLEX$^2$ | 0.326 ±0.016 | **0.102 ±0.010** | 0.338 ±0.020 | **0.278 ±0.017** | 0.104 ±0.010 | **0.034 ±0.007** |

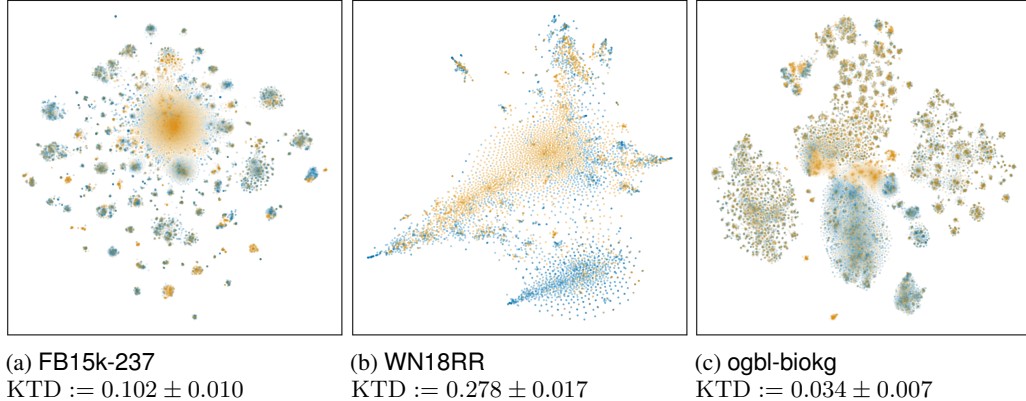

(a) FB15k-237
KTD := $0.102 \pm 0.010$

(b) WN18RR
KTD := $0.278 \pm 0.017$

(c) ogbl-biokg
KTD := $0.034 \pm 0.007$

Figure F.1: **Sampled triples are *close* to test triples.** t-SNE [69] visualisations of the embeddings of test triples (in blue) and triples sampled by COMPLEX$^2$ (in orange). The distribution shift between training and test triples on WN18RR mentioned in Section 7.3 is further confirmed in Fig. F.1b, as it shows a region of test triples (at the bottom and in blue) that is not covered by many generated triples.

### F.5.3 Calibration Diagrams

Existing works on studying the calibration of KGE models are based on interpreting each possible triple $(s, r, o)$ as an independent Bernoulli random variable $Y_{sro}$ whose likelihood is determined by the score function $\phi$, i.e., $\Pr(Y_{sro} = 1 \mid s, r, o) = \sigma(\phi(s, r, o))$ [61, 54, 79], where $\sigma$ denotes the logistic function. While GeKCs encode a probability distribution over all possible triples, this does not impede us to reinterpret them to model the likelihood of each $Y_{sro}$ by still considering scores in log-space as negated energies (see Section 2). Therefore, to evaluate the calibration of GeKCs encoding a non-negative score function $\phi_{\mathsf{pc}}$ (see Section 4) we compute the probability of a triple $(s, r, o)$ being true as $p(Y_{sro} = 1 \mid s, r, o) := \sigma(\log \phi_{\mathsf{pc}}(s, r, o))$. However, the usage of the logistic

function might give misleading results in case of scores not being centred around zero on average. Therefore, we also report calibration diagrams (see paragraph below) where the $p(Y_{sro} = 1 \mid s, r, o)$ is obtained via min-max normalisation of the scores given by KGE models (the logarithm of the scores given for GeKCs), where the minimum and maximum are computed on the training triples. Note that several ex-post (re-)calibration techniques are available [61, 79], but they should benefit GeKCs as they do with existing KGE models.

**Setting and metrics.** To plot calibration diagrams, we follow Socher et al. [59] and sample challenging negative triples, i.e., for each test triple $(s, r, o)$ we sample an unobserved perturbed one $(s, r, \widehat{o})$ by replacing the object with an entity that has appeared at least once with the predicate $r$ in the training data. We then compute the *empirical calibration error* (ECE) [79] as $\text{ECE} := \frac{1}{b} \sum_{i=1}^{b} |p_j - f_j|$, where $b$ is the number of uniformly-chosen bins for the interval $[0, 1]$ of triple probabilities, and $p_j, f_j$ are respectively the average probability and relative frequency of actually existing triples in the $j$-th bin. The lower the ECE score, the better calibrated are the predictions, as they are closer to the empirical frequency of triples that do exist in each bin. The calibration curves are plotted by considering the relative frequency of existing triples in each bin, and curves closer to the main diagonal indicate better calibrated predictions.

**GeKCs are more calibrated out-of-the-box.** Fig. F.2 (resp. Fig. F.3) show calibration diagrams for GeKCs derived from CP and COMPLEX trained with the MLE objective (Eq. (2)) (resp. PLL objective (Eq. (1))). In 19 cases over 24, GeKCs obtained via squaring (Section 4.2) achieve lower ECE scores and better calibrated curves than CP and COMPLEX. While GeKCs obtained via non-negative restriction (Section 4.1) are not well calibrated when using the logistic function, on ogbl-biokg [32] they are still better calibrated than CP and COMPLEX when probabilities are obtained via min-max normalisation. Furthermore, on WN18RR GeKCs achieved the highest ECE scores (corresponding to poorly-calibrated predictions), which could be explained by the distribution shift between training and test triples that was observed for this KG in Section 7.3 and further confirmed in Appendix F.5.2.

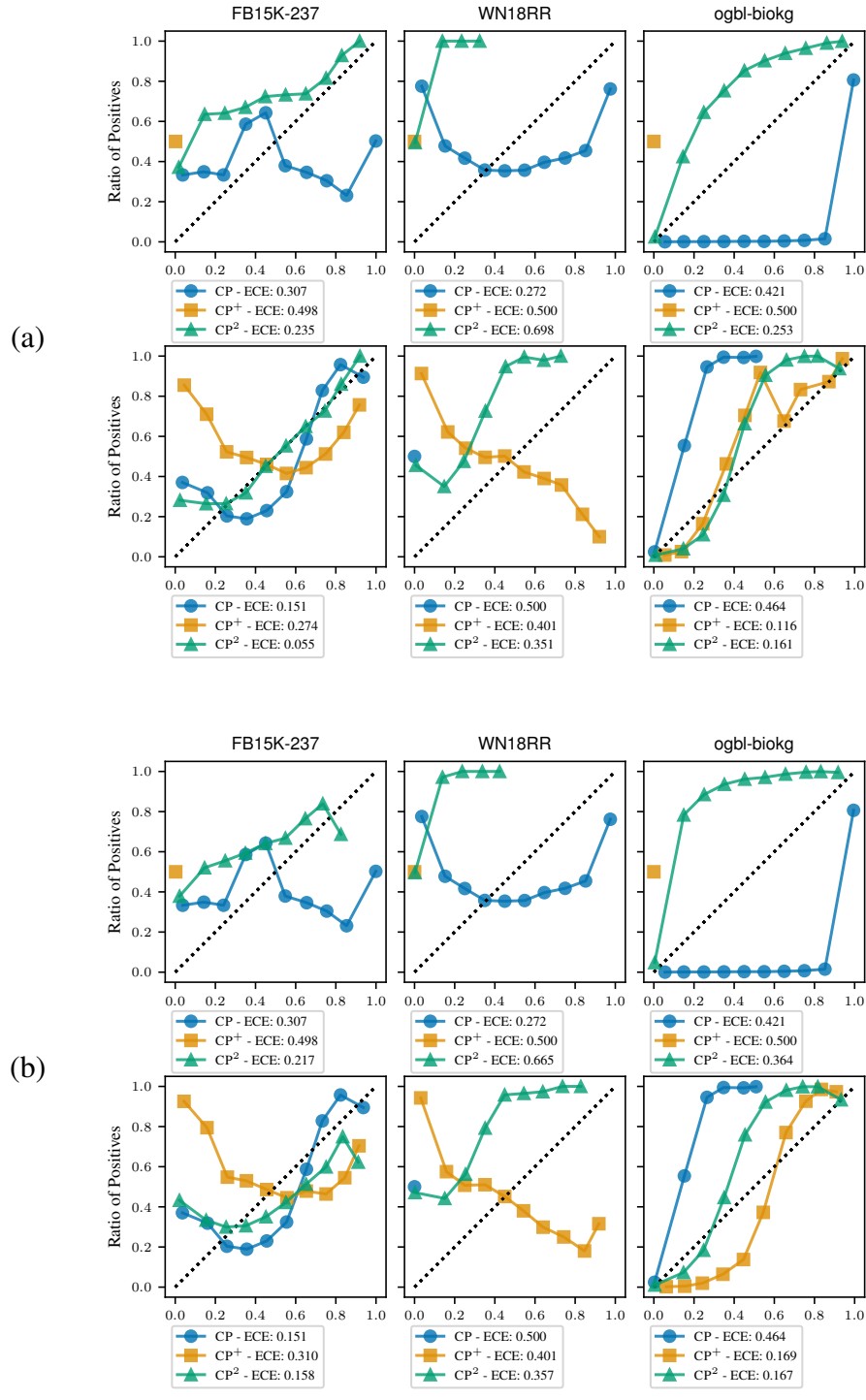

Figure F.2: **Better calibrated predictions with $CP^2$.** Calibration diagrams of CP, $CP^+$ and $CP^2$ trained with either the PLL (Fig. F.2a) or MLE (Fig. F.2b) objectives. The probability of triples are obtained via the application of the logistic function (rows above) and min-max normalisation (rows below). See Appendix F.5.3 for details. The calibration curves for $CP^+$ where triple probabilities are obtained with the logistic function do not provide any meaningful information, as the logarithm of their scores are generally distributed over large negative values.

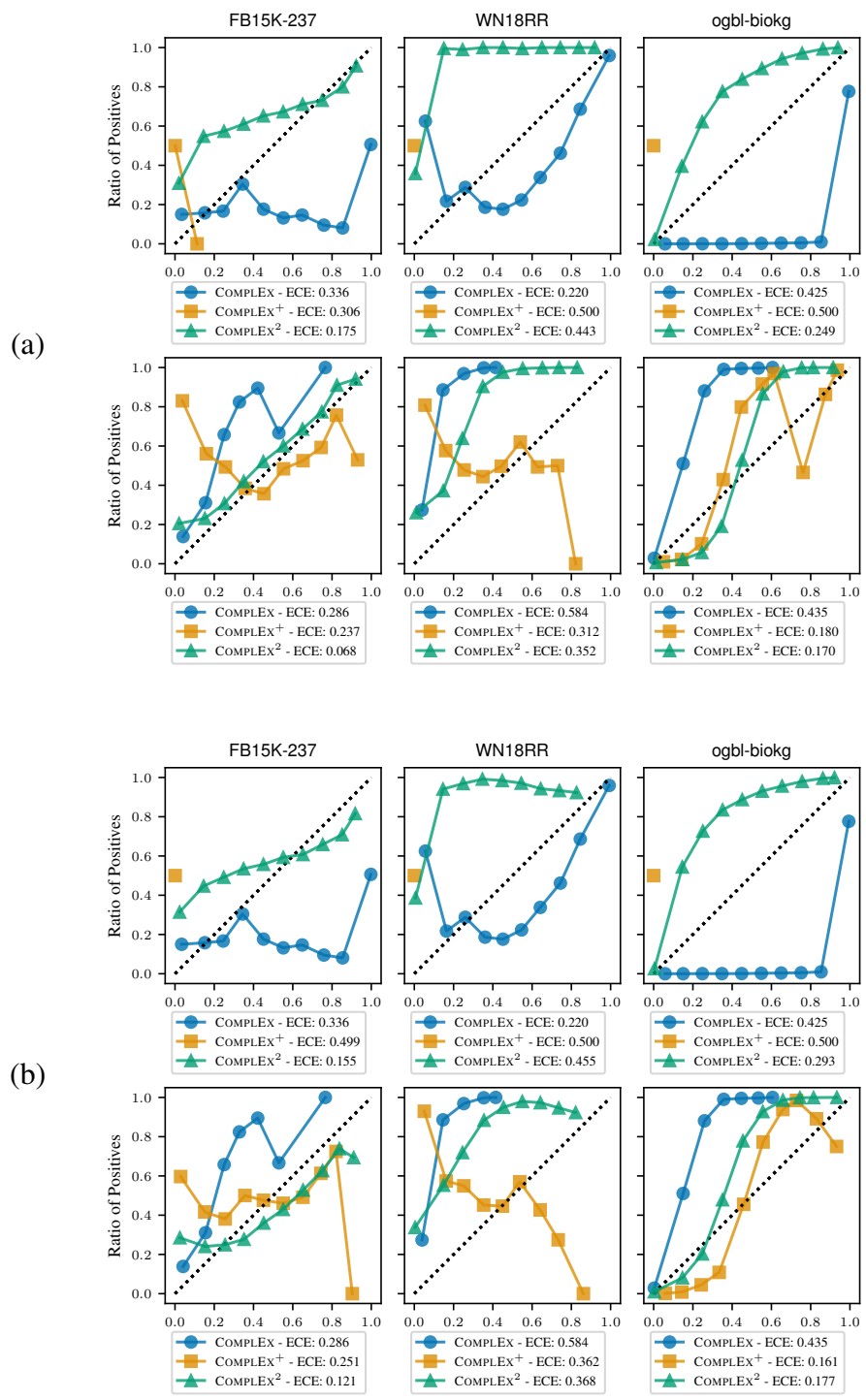

Figure F.3: **Better calibrated predictions with COMPLEX².** Calibration diagrams of COMPLEX, COMPLEX⁺ and COMPLEX² trained with either the PLL (Fig. F.3a) or MLE (Fig. F.3b) objectives. The probability of triples are obtained via the application of the logistic function (rows above) and min-max normalisation (rows below). See Appendix F.5.3 for details. The calibration curves for COMPLEX⁺ where triple probabilities are obtained with the logistic function do not provide any meaningful information, as the logarithm of their scores are generally distributed over large negative values.

