# OpenReview forum: "How to Turn Your Knowledge Graph Embeddings into Generative Models"
_NeurIPS.cc/2023/Conference — NeurIPS 2023 oral_

### Official Review · Reviewer_S5Mx · 2023-07-04

**Soundness:** 3 good
**Presentation:** 3 good
**Contribution:** 3 good
**Rating:** 7
**Confidence:** 3

**Summary:**

This paper provides an new perspective for looking at the popular knowledge graph embedding (KGE) method for knowledge graph completion. Specifically, KGEs models are re-interpreted as circuits. The paper first prove the circuits-based equivalence of score functions of several KGE models, and, in order to regulate KGE further as probabilisitic circuits (GeKC), two restrictions (i.e., non-negative, squaring) are introduced to modify the score functions. Additionally, two techniques are proposed to improve GeKCs: firstly, utilizing trained KGEs as initialization for efficient training of GeKCs, and secondly, introducing constraints to modify the support of GeKCs.

The empirical experiments demonstrate that GeKCs exhibit competitive performance compared to PLL objective applied in KGE. Moreover, the proposed constraint method is justified to avoid constraint violations during inference. Finally, the paper compares GeKCs' ability to the uniform and NNMFAug baseline, showing GeKCs' strong capability to generate new likely triples.

**Strengths:**

Generally the paper presents a novel perspective for the knowledge graph embedding model. This perspective enables researchers to learn KGE models through MLE and therefore build a generative model based on those models. Additionally, logical constraints can be incorporated to this framework to ensure that projected triples are valid during inference.

**Weaknesses:**

The perspective of probabilistic circuits seems to be limited. As in the paper, the score functions of KGE models interpreted as circuits are mainly basd on tensor factorization.

**Questions:**

1. Could other KGE models such as TransE, RotatE be viewed as circuits?
2. In line251, how is |c_k| be calculated? Does that mean constrained GeKCs generally have higher computational cost compared to the one without constraints?
3. Why does training time for squared-COMPLEX in ogbl-biokg decrease compared to COMPLEX/COMPLEX+? Theorem 1 suggests that squared-GeKCs have a higher level of complexity than that indicated in proposition 2.

**Limitations:**

I do not find out any potential negative societal impact.

---

> ### Author Rebuttal · Authors · 2023-08-09
>
> We thank the reviewer for the positive feedback and pointing out how GeKCs are indeed competitive with existing KGE models.
>
> > Could other KGE models such as TransE, RotatE be viewed as circuits?
>
> This is a super interesting question. Here are some ideas.
>
> To our knowledge, distance-based KGE models like TransE and RotatE have not been trained as energy-based models using the pseudo-log-likelihood (PLL). In fact, they are not shown in Figure 2 in [55], which compares several KGE models on different experimental settings.
>
> This is likely because they have different semantics than tensor factorization-based models.  Indeed, the scores computed by TransE and RotatE are “distances”, while scores given by tensor factorization-based models are “similarities”. The first have been interpreted as energies while the latter as negative energies (see L68-70).
>
> One possible way to represent TransE as a probabilistic circuit (PC) is by using a squared norm-2 as distance measure. By doing so, the scores (distances) would be a sum over squares, which can be written as a sum of decomposable squared circuits as done in our work. Representing RotatE as a PC is more challenging. This is because to our knowledge, RotatE is based on a norm-1 on complex embeddings and therefore it would require computing square roots. However, if we slightly modify the original RotatE formulation to use a squared norm-2 instead, we should recover a circuit akin to the one for TransE.
>
> Once we turn TransE/RotatE into a PC, then it would encode a probability distribution $q(S, R, O)$ assigning low probability to likely triples and high probability to unlikely triples, since scores are energies (see above). To solve this issue, it might be possible to model the distribution $p(S,R,O) \propto 1 - q(S, R, O)$ with a circuit and then re-normalizing it (which can be done efficiently). By doing so, we should possibly be able to train them using the PLL objective, akin to the proposed GeKCs. This is certainly an interesting perspective that deserves to be investigated rigorously in a follow up work.
>
>
> > In line251, how is |c_k| be calculated?
>
> $|c_K|$ is calculated by counting the number of edges in the compiled logic circuit, which depends on how complicated the logical formula $K$ is.
>
> For domain constraints, we have shown in Proposition A.1 that in the best case it is linear in the sum of the number of entities $|E|$ and the number of predicates $|R|$. In the worst case is instead linear in the product $|E| \cdot |R|$. In practice, we noticed that on ogbl-biokg the domain constraints are much closer to the best case scenario. In fact, the size of the logic circuit is $306,841$ while $|E|=93,773$ and $|R|=55$.
>
>
> > Does that mean constrained GeKCs generally have higher computational cost compared to the one without constraints?
>
> The computational cost of constrained GeKCs is similar to the cost of multiplying two circuits. In the **worst** case is the product of the sizes of the two circuits, but this is a loose bound and in practice the complexity is much lower as already noted in [D] [72].
>
> Figure 3 supports this claim in practice, since ComplEx^2 brings only a small overhead with respect to ComplEx+ and can be much faster than ComplEx.
>
> To put this in context of the constrained circuits, multiplying by a $c_K$ is going to add a similar overhead, worst case proportional to $|c_K|$. Luckily, the size of $c_K$ for ogbl-biokg is $306,841$. Now consider that the size of the circuit for ComplEx(+) with embedding size 1000 is already much larger, about $375 \cdot 10^6$ and by the same argument in Figure 3 their product should not add more overhead.
>
>
> > Why does training time for squared-COMPLEX in ogbl-biokg decrease compared to COMPLEX/COMPLEX+?
>
> The times shown in Table 1 are relative to the best models found based on validation data, which might have different hyper-parameters and therefore converge at different rates. The results of a more accurate benchmark on the much larger knowledge graph ogbl-wikikg2 are shown in Figure 3.
>
> [D] Y. Shen et al., Tractable Operations for Arithmetic Circuits of Probabilistic Models (2016)

---

### Official Review · Reviewer_t9NK · 2023-07-07

**Soundness:** 3 good
**Presentation:** 4 excellent
**Contribution:** 4 excellent
**Rating:** 7
**Confidence:** 4

**Summary:**

The authors show that standard KGE models, such as COMPLEX, can be represented as structured computational graphs called circuits. They propose a new interpretation of these graphs and their parameters to create efficient and expressive probabilistic models over triples in a KG, called generative KGE circuits (GeKCs). These can be efficiently trained by exact MLE and scale well on large KGs with millions of entities. GeKCs can also sample new triples exactly and efficiently, and predictions at test time never violate logical constraints such as domain schema definitions. Experiments show that these advantages come with little or no loss in link prediction accuracy.

**Strengths:**

- This paper is well-written, with clear language and a well-structured presentation. I enjoy reading it.
- The idea of making KGEs generative is brilliant! There have been so many new KGEs in the past years, we constantly rely on real world datasets. Now with the proposed method, we can actually generate KG datasets conveniently.
- The re-interpretation of existing training objectives as MLE is very clever.
- Additionally, the proposed method makes it easier to add constraints to KGEs.
- The experiments are clearly designed and well-executed, further demonstrating the value of this work.

Overall, this paper is a valuable contribution to the field and deserves a wide audience for its insights and rigorous analysis.

**Weaknesses:**

Maybe not a weakness but some extension?

- The constraint adding part can be extended into another work. I can imagine, in high-stake decision-making applications like finance etc, adding constraints can play a vital role.

**Questions:**

Q1: It is not clear from line 89 why (2) is considered a proxy for (1). Can you provide more details on the reasoning behind this?

Q2: In Table 2, it is not stated why the results from [12] were not included as a baseline. Is there a specific reason for this?

**Limitations:**

Limitations are discussed in sec 7.1 and sec 8.

---

> ### Author Rebuttal · Authors · 2023-08-09
>
> We are grateful to the reviewer for valuing the significance of our contribution as well as its execution.
>
>
> > “The constraint adding part can be extended into another work”
>
> Thanks for recognizing the significance of our work for reliable decision making in safety-critical scenarios. We have some applications of KGE models in biomedical domains in mind, where indeed making trustworthy predictions is crucial if we want to deploy these models and use them. We are very open to additional suggestions.
>
>
> > It is not clear from line 89 why (2) is considered a proxy for (1).
>
> We say the PLL is a proxy of the actual log-likelihood because, under certain assumptions, it is possible to show that the PLL objective can retrieve the MLE solution asymptotically with enough data [C].
>
> To give more context, computing the partition function is infeasible for traditional KGE models when dealing with large knowledge graphs. This generally holds for arbitrary energy-based models (EBMs). In the EBM literature, the pseudo-log-likelihood (PLL) circumvents the problem of computing the log-likelihood exactly as it is easier to compute (it requires summing over only one variable at a time).
>
> An additional difference is that the PLL is a classical discriminative objective while the MLE is a generative one. As such, it permits to obtain a better estimate of the joint probability distribution, and not just (one or many) conditionals, thus being more suitable for generative tasks such as sampling or imputation.
>
>
> > In Table 2, it is not stated why the results from [12] were not included as a baseline.
>
> Our PLL objective is a generalization of the loss used in [12]. In fact, the loss used in [12] can be retrieved from the PLL by setting $\omega_s = \omega_o = 1$ and leaving $\omega_r$ as an hyperparameter. We highlight this in L920-921 in the appendix.
>
> Therefore, our experimental setting with CP and ComplEx when using the PLL encompasses the one in [12] under the column labeled with PLL in Table 1 (here we are assuming the reviewer is referring to Table 1 and not 2, please let us know if that is not the case). The difference in terms of MRR w.r.t. [12] is due to a simplified experimental protocol: a smaller grid search and fixing $\omega_r = 1$. This is because of our limited computational resources, but searching more hyperparameters (and for longer) would likely bring better results for all models.
>
> [C] A. Hyvärinen, Consistency of Pseudolikelihood Estimation of Fully Visible Boltzmann Machines (2006)

---

> > ### Comment · Reviewer_t9NK · 2023-08-14
> >
> > Thank you for your response. I believe that the paper is worth a clear acceptance.

---

### Official Review · Reviewer_P87h · 2023-07-09

**Soundness:** 3 good
**Presentation:** 4 excellent
**Contribution:** 3 good
**Rating:** 7
**Confidence:** 4

**Summary:**

This paper converts score functions of knowledge graph embeddings into generative KGE circuits (GeKCs) by restricting activations to be non-negative or squaring their outputs.
The changes come with little or no loss of link prediction performance,
while enabling probabilistic interpretations, exact maximum-likelihood estimation, better scaling when trained with discriminative objectives, efficient sampling of new triples, and logical constraints such as domain schema definitions.
The paper additionally proposes a metric for measuring the quality of sampled subject-predicate-object triples.

**Strengths:**

- Novel combination of probabilistic circuits and knowledge graph embeddings.
- Clear presentation. Solid theory.

**Weaknesses:**

1. Multiple questions regarding the motivation (i.e. why the introduced properties are useful) need to be clarified. See Questions.
2. Empirical results are not motivating enough.
(a) Sec. 7.1 doesn't show evidence that GeKCs are always preferred for link prediction in terms of accuracy or provide actionable knowledge about on what datasets GeKCs may perform better.
(b) In Sec. 7.2 Integrating Domain Constraints, d-ComplEx2 results are 100 by design, but the baseline ComplEx also achieves very high numbers (99+) while ComplEx2 has worse-than-baseline results. Can you provide d-ComplEx2's general link prediction performance as compared to methods in Sec. 7.1's experiments, so that people can decide whether it is worth using in practice? Do you have results on more than one dataset and more than one model?

**Questions:**

1. To confirm understanding: the model difference between GeKCs and other KGEs is constraining scores to be non-negative or squaring them, and nothing else, right?
2. About scaling.
(a) Why does the GeKCs scale better, given the additional squaring operations or non-negative restrictions?
(b) Is MLE intractable for the models in the paper that can be but are not yet converted to GeKCs? Can you elaborate on why?
3. About loss. When is PLL training recommended and when is MLE training recommended? Why?
4. About probabilities. (a) L29 says the scores cannot be compared across different queries, but L27 gives an example of scores being compared. Why? At least within the same model, negative energies can be compared, right? (b) L29: What are the cases where we want to compare scores across KGE models? (c) What's the benefit of having a probabilistic model instead of an energy based model?
5. About sampling. Why would people want to generate a surrogate KG or do data augmentation? The aim for knowledge graph embeddings is to answer queries over incomplete knowledge graphs. Does having a generative model help this purpose?
6. About logical constraints. Can we set energies to infinity if a query violates logical constraints? It seems that without the probabilistic framework presented in this paper, one may still be able to enforce logical constraints.
7. About generality of the method. This paper turns the score functions of KGEs into probabilistic circuits. Can the same be applied to turn score functions in all sorts of models into PCs?
8. L231: What does consistent mean?
9. Eq. (3): There are two levels of "=". Please differentiate them to avoid difficulty in parsing the formula.
10. L307: Initializing by CP and ComplEx should not be called distillation.
11. L361: How can KGEs be smaller, now that the presented circuits are already with only 2 layers?

Update: The authors addressed my major concerns in the rebuttal.

**Limitations:**

The paper mentions future work which may be considered limitations. It does not seem to discuss negative societal impact, which it may not directly have.

---

> ### Author Rebuttal · Authors · 2023-08-09
>
> We thank the reviewer for all the comments. We believe we addressed all the concerns and are happy to follow up in the discussion phase.
> >The difference between GeKCs and other KGEs is constraining scores or squaring?
>
> Yes, although apparently simple, this modeling difference enables the semantic of normalized probabilities which unlocks a new set of capabilities for KGE models: (i) scaling training to very large KGs (ii) guaranteeing satisfaction of constraints by design which is crucial in high-stake scenarios, and (iii) generative capabilities. This is possible only thanks to recent theoretical advancements in circuits [72].
> >Empirical results are not motivating enough
>
> We remark that the goal of our paper is *not* doing SOTA results for link prediction (L289-290) but to show that a new class of KGEs can excel in other important directions, namely (i-iii) highlighted above. Furthermore, in terms of link prediction only, GeKCs score comparable or better than KGE baselines. Therefore making a strong alternative if one is interested in (i-iii).
> >d-ComplEx2 results are 100 by design, but ComplEx also achieves high numbers
>
> Note that while ComplEx can achieve 99+% semantic consistency, it still predicts more than 360 of semantically invalid triples (see L327-328). Even if it achieved 100% on the test set, it would not guarantee that predictions will be valid for future queries. Our method instead guarantees predictions that will **always be valid**, which is crucial in high-stake applications as pointed out by reviewer t9NK.
> >Can you provide d-ComplEx2's link prediction performance
>
> We did not run D-ComplEx^2 on FB15K-237 and WN18RR because they do not come with constraints, differently from ogbl-biokg. In the attached pdf, we show that GeKCs with domain constraints perform similarly or better than GeKCs alone (and better than other KGEs). We are happy to experiment on additional KGs with domain constraints suggested by the reviewer.
> >Why does the GeKCs scale better?
>
> Because summations in PLL and MLE can be pushed down to the inputs of circuits, thus requiring only linear time w.r.t. the number of entities and predicates (L148-155). This further simplifies the complexity w.r.t. the batch size, as we formalize in Appendix C.4.1-2.
>
> Squared GeKCs adds a minor overhead (see answer 3 to reviewer S5Mx), which is absorbed by the aspects above.
> >When is PLL/MLE recommended?
>
> MLE is recommended to learn better generative models, to improve sample quality (see Table 2). By contrast, as expected by training with a discriminative objective, training via PLL favors link prediction scores, but falls short in other inference tasks. We detail the PLL-MLE relationship in answer 2 to reviewer t9NK.
> >Can Negative energies be compared?
>
> We can only compare the scores of triples that answer the same query to get an ordering for ranking (L27). Scores associated to different queries (eg, different predicates) can differ a lot and are not informative without renormalizing. This can negatively affect a concrete scenario such as complex query reasoning, where scores associated to triples with different predicates are compared. Recently this particular aspect has been investigated in [B].
> >Why comparing scores across KGE models?
>
> Comparing scores is useful to perform model selection or model integration. For model selection, the log-likelihood is the canonical way to compare probabilistic models but requires normalized probabilities. For integration, consider this real-world application: consolidate several existing KGEs from Wikipedia in different languages or integrate them with a probabilistic LLM: we need scores that are normalized in the same range.
> >Why sampling?
>
> Sampling enables us to approximate complex probabilistic inference scenarios, eg, via Monte Carlo estimates. Particularly for KGs, this was useful to approximate the proposed Kernel Triple Distance (Appendix F.3) or to answer complex queries [10,15]. Also, [9] has shown that performing data augmentation via sampling during training can be beneficial for link prediction.
> >Can we set energies to infinity if a query violates constraints?
>
> Filtering EBMs predictions ex-post can be done, but with an external reasoner and this can be very expensive in practice.  Eg, to answer a query $(s,p,?)$ we need first to find all objects $o’$ for which $(s,p,o’)$ violates the constraint. This requires calling a reasoner for possibly millions of objects. For more complex reasoning scenarios, involving predicting multiple links, the number of calls quickly grows (eg, finding all objects $o’$ such that $\exists V.(s,p,V)\land (V,p’,o’)$ is satisfied). The missed opportunity here is to amortize all the intermediate reasoning computations done for the same $(s,p)$, or $(p,o)$ when later we want to answer $(?,p,o)$.
> The knowledge compilation step in our models (L242-244) does exactly this, converting these reasoner calls into a compact circuit structure [18,51].
> >Can we turn all sorts of models into PCs?
>
> Investigating how to turn other KGE models into GeKCs is surely interesting and worth a follow up, please see our responses to reviewers v5Gr, S5Mx.
> >What does consistent mean?
>
> We will specify in the camera-ready that consistency means that entities satisfy the logical constraint.
> >There are two levels of "=".
>
> Thanks, we will fix it in the camera-ready.
> >Initializing by CP and ComplEx should not be called distillation.
>
> Thanks, we will fix that line by mentioning we are indeed performing fine-tuning as reported in Table F.5.
> >How can KGEs be smaller, now that the presented circuits are already with only 2 layers?
>
> We can leverage the data to learn __sparser__ structures of KGE models. For example, by clustering triples (or entities/predicates) we can smartly construct connections encoding particular dependencies and avoid “fully connected” layers as they are now.
>
> [B] E. Arakelyan et al. Adapting Neural Link Predictors for Data-Efficient Complex Query Answering 2023

---

> > ### Comment · Reviewer_P87h · 2023-08-13
> >
> > Thank you for the detailed answers. They addressed my concerns and I have raised my score.

---

> ### Comment · Area_Chair_8hWq · 2023-08-13
> **Please read and respond to the rebuttal**
>
> Dear reviewer,
>
> You rated the paper with the lowest score. Please read and respond to the rebuttal. Has it changed your assessment? Please also read the other reviews which are more positive about the paper.
>
> Thanks!

---

### Official Review · Reviewer_v5Gr · 2023-07-09

**Soundness:** 4 excellent
**Presentation:** 4 excellent
**Contribution:** 4 excellent
**Rating:** 8
**Confidence:** 4

**Summary:**

This paper provides a reduction from several well-known knowledge graph embedding (KGE) methods (e.g., TuckER, CP, etc.) to *decomposable* circuits $\phi$ which can be efficiently computed in $O( (|E| + |R|) |\phi|)$ (note that this is mostly due to the decomposable nature of TuckER and its derivatives). The authors then address the problem that $\phi$ can be negative and hence suitable to directly represent a probability distribution by applying one of two operations---non-negativity or squaring---which allows the circuit to be efficiently learnt and sampled from. Finally, they show how the circuit formalism allows one to easily apply domain constraints as the product of the two circuits.

The authors then evaluate their proposed variants of the KGE methods on four knowledge graphs, where they show (1) their squared variant produces comparable or better results to the original variants with significant computational speedups, (2) their method guarantees domain constraints, and (3) their squared method generates higher quality samples than state-of-the-art. They authors present several additional results in their appendices including: showing that it is feasible to distilling parameters from already trained circuits to the proposed variants (which allows them to be sampled from) and showing that the proposed variants are better calibrated than the original ones.

**Strengths:**

* This is a meticulously executed paper with thorough empirical evaluation. While its key results are in the paper, there are many more interesting and insightful experiments in the appendix.
* The paper presents a useful and elegant theoretical framework for computing knowledge graph embeddings that *extends* existing methods like TuckER or ComplEx allowing them to be learned and sampled from efficiently (it's not just conceptual).
* The paper is extremely well-written and clear.

**Weaknesses:**

* While the paper reduces several existing KGE methods to circuits, the reduction is quite straightforward and only practical for this family of circuits. There are other KGE methods (e.g. graph neural networks) that are not covered by this paper or its related work.

**Questions:**

The paper is extremely well-written and clear, and I don't have any questions.

**Limitations:**

* While the paper reduces several existing KGE methods to circuits, the reduction is quite straightforward and only practical for this family of circuits.

---

> ### Author Rebuttal · Authors · 2023-08-09
>
> We thank the reviewer for appreciating the rigor, novelty and direction of our research! Such a simple idea can open up many valuable perspectives.
>
> > There are other KGE methods (e.g. graph neural networks) that are not covered by this paper.
>
> Indeed, connecting circuits with more general KGEs such as GNNs is a promising idea.  Unfortunately, these models are generally more difficult to deal with, given the presence of some non-linearities that cannot be modeled within the language of tractable circuits. However, recently a connection between GNNs and tensor factorization-based KGE models has been established [A], thus our analysis might well be useful  for connecting GNN-based models and circuits in future work.
>
> Additionally, in our response 1 to reviewer S5Mx we sketch how to turn other KGE models such as TransE and RotatE into GeKCs. However, this is not straightforward and is left to future  work.
>
> [A] Y. Chen et al. ReFactor GNNs: Revisiting Factorisation-based Models from a Message-Passing Perspective (2022).

---

### Author Rebuttal · Authors · 2023-08-09

We thank all reviewers for the time spent reviewing the paper and recognizing the **significance** (“useful and elegant theoretical framework” – v5Gr, “solid theory” – P87h, “deserves a wide audience for its insights and rigorous analysis” – t9NK) and **novelty** (“Novel combination” – P87h, “making KGEs generative is brilliant!” – t9NK, “novel perspective” – S5Mx) of our research direction, as well as the **quality of the presentation** (“meticulously executed paper” – v5Gr, “clear presentation” – P87h, “well-written, with clear language and a well-structured presentation” – t9NK).

We provided below individual answers to your reviews, and we are happy to follow up on any aspect in the discussion phase.

---

### Decision · Program_Chairs · 2023-09-21

**Decision:**

Accept (oral)

**Comment:**

The paper introduces an innovative perspective on knowledge graph embedding (KGE) models, allowing for generative model construction through maximum likelihood estimation (MLE). The incorporation of logical constraints enhances inference validity. The paper's clarity, well-structured presentation, and reimagining of training objectives as MLE have been mentioned as its strengths. The well-designed experiments further support the proposed methods. All reviewers are positive to very positive about his submission. I strongly recommend accepting this paper, as it significantly contributes a fresh perspective to the evolving landscape of KGE research.